# Trust Region-Based Safe Distributional Reinforcement Learning for Multiple Constraints

**Dohyeong Kim**[1]**, Kyungjae Lee**[2]**, and Songhwai Oh**[1]

[1]Dep. of Electrical and Computer Engineering and ASRI, Seoul National University
[2]Artificial Intelligence Graduate School, Chung-Ang University
dohyeong.kim@rllab.snu.ac.kr, kyungjae.lee@ai.cau.ac.kr,
songhwai@snu.ac.kr

## Abstract

In safety-critical robotic tasks, potential failures must be reduced, and multiple constraints must be met, such as avoiding collisions, limiting energy consumption, and maintaining balance. Thus, applying safe reinforcement learning (RL) in such robotic tasks requires to handle multiple constraints and use risk-averse constraints rather than risk-neutral constraints. To this end, we propose a trust region-based safe RL algorithm for multiple constraints called a *safe distributional actor-critic* (SDAC). Our main contributions are as follows: 1) introducing a gradient integration method to manage infeasibility issues in multi-constrained problems, ensuring theoretical convergence, and 2) developing a TD($\lambda$) target distribution to estimate risk-averse constraints with low biases. We evaluate SDAC through extensive experiments involving multi- and single-constrained robotic tasks. While maintaining high scores, SDAC shows 1.93 times fewer steps to satisfy all constraints in multi-constrained tasks and 1.78 times fewer constraint violations in single-constrained tasks compared to safe RL baselines. Code is available at: https://github.com/rllab-snu/Safe-Distributional-Actor-Critic.

## 1 Introduction

Deep reinforcement learning (RL) enables reliable control of complex robots [Merel et al., 2020, Peng et al., 2021, Rudin et al., 2022]. In order to successfully apply RL to real-world systems, it is essential to design a proper reward function which reflects safety guidelines, such as collision avoidance and limited energy consumption, as well as the goal of the given task. However, finding the reward function that considers all such factors involves a cumbersome and time-consuming process since RL algorithms must be repeatedly performed to verify the results of the designed reward function. Instead, *safe RL*, which handles safety guidelines as constraints, is an appropriate solution. A safe RL problem can be formulated using a constrained Markov decision process [Altman, 1999], where not only the reward but also cost functions are defined to provide the safety guideline signals. By defining constraints using expectation or risk measures of the sum of costs, safe RL aims to maximize returns while satisfying the constraints. Under the safe RL framework, the training process becomes straightforward since there is no need to search for a reward that reflects the safety guidelines.

While various safe RL algorithms have been proposed to deal with the safety guidelines, their applicability to general robotic applications remains limited due to the insufficiency in **1)** handling multiple constraints and **2)** minimizing failures, such as robot breakdowns after collisions. First, many safety-critical applications require multiple constraints, such as maintaining distance from obstacles, limiting operational space, and preventing falls. Lagrange-based safe RL methods [Yang et al., 2021, Zhang and Weng, 2022, Bai et al., 2022], which convert a safe RL problem into a dual problem and

update the policy and Lagrange multipliers, are commonly used to solve these multi-constrained problems. However, the Lagrangian methods are difficult to guarantee satisfying constraints during training theoretically, and the training process can be unstable due to the multipliers [Stooke et al., 2020]. To this end, trust region-based methods [Yang et al., 2020, Kim and Oh, 2022a], which can ensure to improve returns while satisfying the constraints under tabular settings [Achiam et al., 2017], have been proposed as an alternative to stabilize the training process. Still, trust region-based methods have a critical issue. Depending on the initial policy settings, there can be an infeasible starting case, meaning that no policy within the trust region satisfies constraints. To address this issue, we can sequentially select a violated constraint and update the policy to reduce the selected constraint [Xu et al., 2021]. However, this can be inefficient as only one constraint is considered per update. It will be better to handle multiple constraints at once, but it is a remaining problem to find a policy gradient that reflects several constraints and guarantees to reach a feasible policy set.

Secondly, as RL settings are inherently stochastic, employing risk-neutral measures like expectation to define constraints can lead to frequent failures. Hence, it is crucial to define constraints using risk measures, such as conditional value at risk (CVaR), as they can reduce the potential for massive cost returns by emphasizing tail distributions [Yang et al., 2021, Kim and Oh, 2022a]. In safe RL, critics are used to estimate the constraint values. Especially, to estimate constraints based on risk measures, it is required to use distributional critics [Dabney et al., 2018b], which can be trained using the distributional Bellman update [Bellemare et al., 2017]. However, the Bellman update only considers the one-step temporal difference, which can induce a large bias. The estimation bias makes it difficult for critics to judge the policy, which can lead to the policy becoming overly conservative or risky, as shown in Section 5.3. In particular, when there are multiple constraints, the likelihood of deriving incorrect policy gradients due to estimation errors grows exponentially. Therefore, there is a need for a method that can train distributional critics with low biases.

In this paper, we propose a trust region-based safe RL algorithm called a *safe distributional actor-critic* (SDAC), designed to effectively manage multiple constraints and estimate risk-averse constraints with low biases. First, to handle the infeasible starting case by considering all constraints simultaneously, we propose a *gradient integration method* that projects unsafe policies into a feasible policy set by solving a quadratic program (QP) consisting of gradients of all constraints. It guarantees to obtain a feasible policy within a finite time under mild technical assumptions, and we experimentally show that it can restore the policy more stably than the existing method [Xu et al., 2021]. Furthermore, by updating the policy using the trust region method with the integrated gradient, our approach makes the training process more stable than the Lagrangian method, as demonstrated in Section 5.2. Second, to train critics to estimate constraints with low biases, we propose a *TD($\lambda$) target distribution* which can adjust the bias-variance trade-off. The target distribution is obtained by merging the quantile regression losses [Dabney et al., 2018b] of multi-step distributions and extracting a unified distribution from the loss. The unified distribution is then projected onto a quantile distribution set in a memory-efficient manner. We experimentally show that the target distribution can trade off the bias-variance of the constraint estimations (see Section 5.3).

We conduct extensive experiments with multi-constrained locomotion tasks and single-constrained Safety Gym tasks [Ray et al., 2019] to evaluate the proposed method. In the locomotion tasks, SDAC shows 1.93 times fewer steps to satisfy all constraints than the second-best baselines. In the Safety Gym tasks, the proposed method shows 1.78 times fewer constraint violations than the second-best methods while achieving high returns when using risk-averse constraints. As a result, it is shown that the proposed method can efficiently handle multiple constraints using the gradient integration method and effectively lower the constraint violations using the low-biased distributional critics.

## 2 Background

**Constrained Markov Decision Processes.** We formulate the safe RL problem using constrained Markov decision processes (CMDPs) [Altman, 1999]. A CMDP is defined as $(S, A, P, R, C_{1,..,K}, \rho, \gamma)$, where $S$ is a state space, $A$ is an action space, $P : S \times A \times S \mapsto [0, 1]$ is a transition model, $R : S \times A \times S \mapsto \mathbb{R}$ is a reward function, $C_{k \in \{1,...,K\}} : S \times A \times S \mapsto \mathbb{R}_{\geq 0}$ are cost functions, $\rho : S \mapsto [0, 1]$ is an initial state distribution, and $\gamma \in (0, 1)$ is a discount factor. Given a policy $\pi$ from a stochastic policy set $\Pi$, the discounted state distribution is defined as $d^\pi(s) := (1 - \gamma) \sum_{t=0}^\infty \gamma^t \Pr(s_t = s | \pi)$, and the return is defined as $Z_R^\pi(s, a) := \sum_{t=0}^\infty \gamma^t R(s_t, a_t, s_{t+1})$, where $s_0 = s$, $a_0 = a$, $a_t \sim \pi(\cdot | s_t)$, and $s_{t+1} \sim P(\cdot | s_t, a_t)$. Then, the state value and state action

value functions are defined as: $V_R^\pi(s) := \mathbb{E}\left[Z_R^\pi(s,a)|a \sim \pi(\cdot|s)\right]$, $Q_R^\pi(s,a) := \mathbb{E}\left[Z_R^\pi(s,a)\right]$. By substituting the costs for the reward, the cost value functions $V_{C_k}^\pi(s)$ and $Q_{C_k}^\pi(s,a)$ are defined. In the remainder of the paper, the cost parts will be omitted since they can be retrieved by replacing the reward with the costs. Then, the safe RL problem is defined as follows with a safety measure $F$:

$$\text{maximize}_\pi \ J(\pi) \ \textbf{s.t.} \ F(Z_{C_k}^\pi(s,a)|s \sim \rho, a \sim \pi(\cdot|s)) \le d_k \ \forall k, \tag{1}$$

where $J(\pi) := \mathbb{E}\left[Z_R^\pi(s,a)|s \sim \rho, a \sim \pi(\cdot|s)\right] + \beta\mathbb{E}\left[\sum_{t=0}^\infty \gamma^t H(\pi(\cdot|s_t))|\rho, \pi, P\right]$, $\beta$ is an entropy coefficient, $H$ is the Shannon entropy, and $d_k$ is a threshold of the $k$th constraint.

**Trust-Region Method With a Mean-Std Constraint.** Kim and Oh [2022a] have proposed a trust region-based safe RL method with a risk-averse constraint, called a *mean-std* constraint. The definition of the mean-std constraint function is as follows:

$$F(Z; \alpha) := \mathbb{E}[Z] + (\phi(\Phi^{-1}(\alpha))/\alpha) \cdot \text{Std}[Z], \tag{2}$$

where $\alpha \in (0, 1]$ adjusts the risk level of constraints, $\text{Std}[Z]$ is the standard deviation of $Z$, and $\phi$ and $\Phi$ are the probability density function and the cumulative distribution function (CDF) of the standard normal distribution, respectively. In particular, setting $\alpha = 1$ causes the standard deviation part to be zero, so the constraint becomes a risk-neutral constraint. Also, the mean-std constraint can effectively reduce the potential for massive cost returns, as shown in Yang et al. [2021], Kim and Oh [2022a,b]. In order to calculate the mean-std constraint, it is essential to estimate the standard deviation of the cost return. To this end, Kim and Oh [2022a] define the square value functions:

$$S_{C_k}^\pi(s) := \mathbb{E}\left[Z_{C_k}^\pi(s,a)^2|a \sim \pi(\cdot|s)\right], \ S_{C_k}^\pi(s,a) := \mathbb{E}\left[Z_{C_k}^\pi(s,a)^2\right]. \tag{3}$$

Since $\text{Std}[Z]^2 = \mathbb{E}[Z^2] - \mathbb{E}[Z]^2$, the $k$th constraint can be written as follows:

$$F_k(\pi; \alpha) = J_{C_k}(\pi) + (\phi(\Phi^{-1}(\alpha))/\alpha) \cdot \sqrt{J_{S_k}(\pi) - J_{C_k}(\pi)^2} \le d_k, \tag{4}$$

where $J_{C_k}(\pi) := \mathbb{E}\left[V_{C_k}^\pi(s)|s \sim \rho\right]$, $J_{S_k}(\pi) := \mathbb{E}\left[S_{C_k}^\pi(s)|s \sim \rho\right]$. In order to apply the trust region method [Schulman et al., 2015], it is necessary to derive surrogate functions for the objective and constraints. These surrogates can substitute for the objective and constraints within the trust region. Given a behavioral policy $\mu$ and the current policy $\pi_{\text{old}}$, we denote the surrogates as $J^{\mu,\pi_{\text{old}}}(\pi)$ and $F_k^{\mu,\pi_{\text{old}}}(\pi; \alpha)$. For the definition and derivation of the surrogates, please refer to Appendix A.7 and [Kim and Oh, 2022a]. Using the surrogates, a policy can be updated by solving the following subproblem:

$$\text{maximize}_{\pi'} \ J^{\mu,\pi}(\pi') \ \textbf{s.t.} \ D_{KL}(\pi||\pi') \le \epsilon, F_k^{\mu,\pi}(\pi', \alpha) \le d_k \ \forall k, \tag{5}$$

where $D_{\text{KL}}(\pi||\pi') := \mathbb{E}_{s \sim d^\mu}\left[D_{\text{KL}}(\pi(\cdot|s)||\pi'(\cdot|s))\right]$, $D_{\text{KL}}$ is the KL divergence, and $\epsilon$ is a trust region size. This subproblem can be solved through approximation and a line search (see Appendix A.8). However, it is possible that there is no policy satisfying the constraints of (5). In order to tackle this issue, the policy must be projected onto a feasible policy set that complies with all constraints, yet there is a lack of such methods. In light of this issue, we introduce an efficient feasibility handling method for multi-constrained RL problems.

**Distributional Quantile Critic.** Dabney et al. [2018b] have proposed a method for approximating the random variable $Z_R^\pi$ to follow a quantile distribution. Given a parametric model, $\theta : S \times A \mapsto \mathbb{R}^M$, $Z_R^\pi$ can be approximated as $Z_{R,\theta}$, called a *distributional quantile critic*. The probability density function of $Z_{R,\theta}$ is defined as follows:

$$\text{Pr}(Z_{R,\theta}(s,a) = z) := \sum_{m=1}^M \delta_{\theta_m(s,a)}(z)/M, \tag{6}$$

where $M$ is the number of atoms, $\theta_m(s,a)$ is the $m$th atom, $\delta$ is the Dirac function, and $\delta_a(z) := \delta(z-a)$. The percentile value of the $m$th atom is denoted by $\tau_m$ ($\tau_0 = 0, \tau_i = i/M$). In distributional RL, the returns are directly estimated to get value functions, and the target distribution can be calculated from the distributional Bellman operator [Bellemare et al., 2017]: $\mathcal{T}^\pi Z_R(s,a) \stackrel{D}{:=} R(s,a,s') + \gamma Z_R(s',a')$, where $s' \sim P(\cdot|s,a)$ and $a' \sim \pi(\cdot|s')$. The above one-step distributional operator can be expanded to the $n$-step one: $\mathcal{T}_n^\pi Z_R(s_0,a_0) \stackrel{D}{:=} \sum_{t=0}^{n-1} \gamma^t R(s_t,a_t,s_{t+1}) + \gamma^n Z_R(s_n,a_n)$, where $a_t \sim \pi(\cdot|s_t)$ for $t = 1,...,n$. Then, the critic can be trained to minimize the following quantile regression loss [Dabney et al., 2018b]:

$$\mathcal{L}(\theta) = \sum_{m=1}^M \underbrace{\mathbb{E}_{(s,a) \sim D}\left[\mathbb{E}_{Z \sim \mathcal{T}^\pi Z_{R,\theta}(s,a)}\left[\rho_{\bar{\tau}_m}(Z - \theta_m(s,a))\right]\right]}_{=:\mathcal{L}_{\text{QR}}^{\bar{\tau}_m}(\theta_m)}, \ \text{where} \ \rho_\tau(x) = x \cdot (\tau - \mathbf{1}_{x<0}), \tag{7}$$

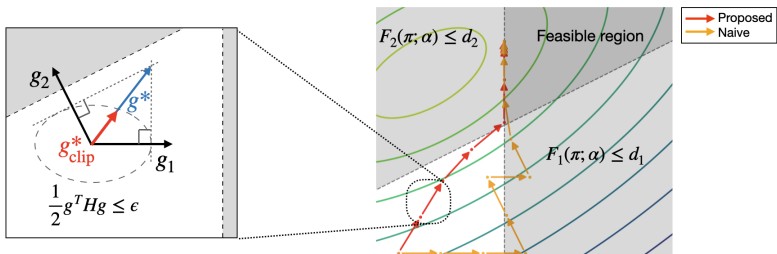

Figure 1: **Left: Gradient integration.** Each constraint is truncated to be tangent to the trust region indicated by the ellipse, and the dashed straight lines show the truncated constraints. The solution of (8) is indicated in blue, pointing to the nearest point in the intersection of the truncated constraints. If the solution crosses the trust region, parameters are updated by the clipped direction, shown in red. **Right: Optimization paths of the proposed and naive method in a toy example.** The description is presented in Appendix A.2. The contour graph represents the objective of the toy example. The optimization paths exhibit distinct characteristics due to the difference that the naive method reflects only one constraint and the proposed method considers all constraints at once.

$D$ is a replay buffer, $\bar{\tau}_m := (\tau_{m-1} + \tau_m)/2$, and $\mathcal{L}_{\mathrm{QR}}^{\bar{\tau}_m}(\theta_m)$ denotes the quantile regression loss for a single atom. The distributional quantile critic can be plugged into existing actor-critic algorithms because only the critic modeling part is changed.

## 3 Proposed Method

The proposed method comprises two key components: **1)** a feasibility handling method required for multi-constrained safe RL problems and **2)** a target distribution designed to minimize estimation bias. This section sequentially presents these components, followed by a detailed explanation of the proposed method.

### 3.1 Feasibility Handling For Multiple Constraints

An optimal safe RL policy can be found by iteratively solving the subproblem (5), but the feasible set of (5) can be empty in the infeasible starting cases. To address the feasibility issue in safe RL with multiple constraints, one of the violated constraints can be selected, and the policy is updated to minimize the constraint until the feasible region is not empty [Xu et al., 2021], which is called a *naive approach*. However, it may not be easy to quickly reach the feasible condition if only one constraint at each update step is used to update the policy. Therefore, we propose a feasibility handling method which reflect all the constraints simultaneously, called a *gradient integration method*. The main idea is to get a gradient that reduces the value of violated constraints and keeps unviolated constraints. To find such a gradient, the following quadratic program (QP) can be formulated by linearly approximating the constraints:

$$g^* = \underset{g}{\mathrm{argmin}} \frac{1}{2} g^T H g \ \ \textbf{s.t.} \ g_k^T g + c_k \leq 0 \ \forall k, \ c_k := \min(\sqrt{2\epsilon g_k^T H^{-1} g_k}, F_k(\pi_\psi; \alpha) - d_k + \zeta), \quad (8)$$

where $H$ is the Hessian of KL divergence between the previous policy and the current policy with parameters $\psi$, $g_k$ is the gradient of the $k$th constraint, $c_k$ is a truncated threshold to make the $k$th constraint tangent to the trust region, $\epsilon$ is a trust region size, and $\zeta \in \mathbb{R}_{>0}$ is a slack coefficient. The reason why we truncate constraints is to make the gradient integration method invariant to the gradient scale. Otherwise, constraints with larger gradient scales might produce a dominant policy gradient. Finally, we update the policy parameters using the clipped gradient as follows:

$$\psi^* = \psi + \min(1, \sqrt{2\epsilon/(g^{*T} H g^*)})g^*. \quad (9)$$

Figure 1 illustrates the proposed gradient integration process. In summary, the policy is updated by solving (5); if there is no solution to (5), it is updated using the gradient integration method. Then, the policy can reach the feasibility condition within finite time steps.

**Theorem 3.1.** *Assume that the constraints are differentiable and convex, gradients of the constraints are L-Lipschitz continuous, eigenvalues of the Hessian are equal or greater than a positive value $R \in \mathbb{R}_{>0}$, and $\{\psi|F_k(\pi_\psi; \alpha) + \zeta < d_k, \ \forall k\} \neq \emptyset$. Then, there exists $E \in \mathbb{R}_{>0}$ such that if*

$0 < \epsilon \leq E$ and a policy is updated by the proposed gradient integration method, all constraints are satisfied within finite time steps.

Note that the first two assumptions of Theorem 3.1 are commonly used in multi-task learning [Liu et al., 2021, Yu et al., 2020, Navon et al., 2022], and the assumption on eigenvalues is used in most trust region-based RL methods [Schulman et al., 2015, Kim and Oh, 2022a], so the assumptions in Theorem 3.1 can be considered reasonable. We provide the proof and show the existence of a solution (8) in Appendix A.1. The provided proof shows that the constant $E$ is proportional to $\zeta$. This means that the trust region size should be set smaller as $\zeta$ decreases. Also, we further analyze the worst-case time to satisfy all constraints by comparing the gradient integration method and naive approach in Appendix A.3. In conclusion, if the policy update rule (5) is not feasible, a finite number of applications of the gradient integration method will make the policy feasible.

### 3.2 TD($\lambda$) Target Distribution

The mean-std constraints can be estimated using the distributional quantile critics. Since the estimated constraints obtained from the critics are directly used to update policies in (5), estimating the constraints with low biases is crucial. In order to reduce the estimation bias of the critics, we propose a target distribution by capturing that the TD($\lambda$) loss, which is obtained by a weighted sum of several losses, and the quantile regression loss with a single distribution are identical. A recursive method is then introduced so that the target distribution can be obtained practically. First, the $n$-step targets for the current policy $\pi$ are estimated as follows, after collecting trajectories $(s_t, a_t, s_{t+1}, ...)$ with a behavioral policy $\mu$:

$$\hat{Z}_R^{(n)}(s_t, a_t) :\overset{D}{=} R_t + \gamma R_{t+1} + \cdots + \gamma^{n-1} R_{t+n-1} + \gamma^n Z_{R,\theta}(s_{t+n}, \hat{a}_{t+n}), \tag{10}$$

where $R_t = R(s_t, a_t, s_{t+1})$, and $\hat{a}_{t+n} \sim \pi(\cdot|s_{t+n})$. Note that the $n$-step target controls the bias-variance tradeoff using $n$. If $n$ is equal to 1, the $n$-step target is equivalent to the temporal difference method that has low variance but high bias. On the contrary, if $n$ increases to infinity, it becomes a Monte Carlo estimation that has high variance but low bias. However, finding proper $n$ is another cumbersome task. To alleviate this issue, TD($\lambda$) [Sutton, 1988] method considers the discounted sum of all $n$-step targets. Similar to TD($\lambda$), we define the TD($\lambda$) loss for the distributional quantile critic as the discounted sum of all quantile regression losses with $n$-step targets. Then, the TD($\lambda$) loss for a single atom is approximated using importance sampling of the sampled $n$-step targets in (10) as:

$$\mathcal{L}_{\mathrm{QR}}^{\bar{\tau}_m}(\theta_m) = (1-\lambda) \sum_{i=0}^{\infty} \lambda^i \mathbb{E}_{(s_t,a_t)\sim D} \left[ \mathbb{E}_{Z \sim \mathcal{T}_{i+1}^\pi Z_{R,\theta}(s_t,a_t)} [\rho_{\bar{\tau}_m}(Z - \theta_m(s_t,a_t))] \right]$$

$$= (1-\lambda) \sum_{i=0}^{\infty} \lambda^i \mathbb{E}_{(s_t,a_t)\sim D} \left[ \mathbb{E}_{Z \sim \mathcal{T}_{i+1}^\mu Z_{R,\theta}(s_t,a_t)} \left[ \prod_{j=t+1}^{t+i} \frac{\pi(a_j|s_j)}{\mu(a_j|s_j)} \rho_{\bar{\tau}_m}(Z - \theta_m(s_t,a_t)) \right] \right] \tag{11}$$

$$\approx (1-\lambda) \sum_{i=0}^{\infty} \lambda^i \mathbb{E}_{(s_t,a_t)\sim D} \left[ \prod_{j=t+1}^{t+i} \frac{\pi(a_j|s_j)}{\mu(a_j|s_j)} \sum_{m=1}^{M} \frac{1}{M} \rho_{\bar{\tau}_m}(\hat{Z}_{R,m}^{(i+1)}(s_t,a_t) - \theta_m(s_t,a_t)) \right],$$

where $\lambda$ is a trace-decay value, and $\hat{Z}_{R,m}^{(i)}$ is the $m$th atom of $\hat{Z}_R^{(i)}$. Since $\hat{Z}_R^{(i)}(s_t, a_t) \overset{D}{=} R_t + \gamma \hat{Z}_R^{(i-1)}(s_{t+1}, a_{t+1})$ is satisfied, (11) is the same as the quantile regression loss with the following single distribution $\hat{Z}_t^{\mathrm{tot}}$, called a *TD($\lambda$) target distribution*:

$$\Pr(\hat{Z}_t^{\mathrm{tot}} = z) := \frac{1}{\mathcal{N}_t}(1-\lambda) \sum_{i=0}^{\infty} \lambda^i \prod_{j=t+1}^{t+i} \frac{\pi(a_j|s_j)}{\mu(a_j|s_j)} \sum_{m=1}^{M} \frac{1}{M} \delta_{\hat{Z}_{R,m}^{(i+1)}(s_t,a_t)}(z)$$

$$= \frac{1-\lambda}{\mathcal{N}_t} \left( \sum_{m=1}^{M} \frac{1}{M} \delta_{\hat{Z}_{t,m}^{(1)}}(z) + \lambda \frac{\pi(a_{t+1}|s_{t+1})}{\mu(a_{t+1}|s_{t+1})} \sum_{i=0}^{\infty} \lambda^i \prod_{j=t+2}^{t+1+i} \frac{\pi(a_j|s_j)}{\mu(a_j|s_j)} \sum_{m=1}^{M} \frac{1}{M} \delta_{\hat{Z}_{R,m}^{(i+2)}(s_t,a_t)}(z) \right) \tag{12}$$

$$= \frac{1-\lambda}{\mathcal{N}_t} \underbrace{\sum_{m=1}^{M} \frac{1}{M} \delta_{\hat{Z}_{t,m}^{(1)}}(z)}_{(a)} + \lambda \frac{\mathcal{N}_{t+1}}{\mathcal{N}_t} \frac{\pi(a_{t+1}|s_{t+1})}{\mu(a_{t+1}|s_{t+1})} \underbrace{\Pr(R_t + \gamma \hat{Z}_{t+1}^{\mathrm{tot}} = z)}_{(b)},$$

where $\mathcal{N}_t$ is a normalization factor. If the target for time step $t+1$ is obtained, the target distribution for time step $t$ becomes the weighted sum of **(a)** the current one-step TD target and **(b)** the shifted

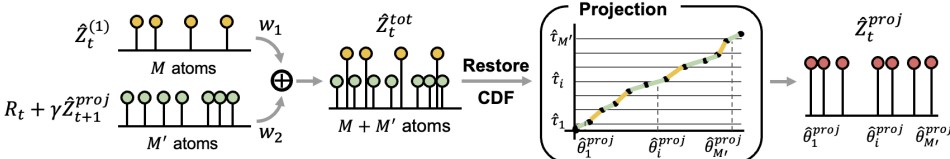

Figure 2: **Procedure for constructing the target distribution.** First, multiply the target at $t+1$ step by $\gamma$ and add $R_t$. Next, weight-combine the shifted previous target and one-step target at $t$ step and restore the CDF of the combined target. The CDF can be restored by sorting the positions of the atoms and then accumulating the weights at each atom position. Finally, the projected target can be obtained by finding the positions of the atoms corresponding to $M'$ quantiles in the CDF. Using the projected target, the target at $t-1$ step can be found recursively.

previous target distribution, so it can be obtained recursively, as shown in (12). Since the definition requires infinite sums, the recursive way is more practical for computing the target. Nevertheless, to obtain the target in that recursive way, we need to store all quantile positions and weights for all time steps, which is not memory-efficient. Therefore, we propose to project the target distribution into a quantile distribution with a specific number of atoms, $M'$ (we set $M' > M$ to reduce information loss). The overall process to get the TD($\lambda$) target distribution is illustrated in Figure 2, and the pseudocode is given in Appendix A.5. Furthermore, we can show that a distribution trained with the proposed target converges to the distribution of $Z_R^\pi$.

**Theorem 3.2.** *Let define a distributional operator* $\mathcal{T}_\lambda^{\mu,\pi}$, *whose probability density function is:*

$$\Pr(\mathcal{T}_\lambda^{\mu,\pi} Z(s,a) = z) \propto$$
$$\sum_{i=0}^\infty \mathbb{E}_\mu \left[ \lambda^i \prod_{j=1}^i \frac{\pi(a_j|s_j)}{\mu(a_j|s_j)} \mathop{\mathbb{E}}_{a' \sim \pi(\cdot|s_{i+1})} \left[ \Pr\left( \sum_{t=0}^i \gamma^t R_t + \gamma^{i+1} Z(s_{i+1}, a') = z \right) \right] \Big| s_0 = s, a_0 = a \right]. \quad (13)$$

*Then, a sequence,* $Z_{k+1}(s,a) = \mathcal{T}_\lambda^{\mu,\pi} Z_k(s,a) \ \forall (s,a)$, *converges to* $Z_R^\pi$.

The TD($\lambda$) target is a quantile distribution version of the distributional operator $\mathcal{T}_\lambda^{\mu,\pi}$ in Theorem 3.2. Consequently, a distribution updated by minimizing the quantile regression loss with the TD($\lambda$) target converges to the distribution of $Z_R^\pi$ if the number of atoms is infinite, according to Theorem 3.2. The proof of Theorem 3.2 is provided in Appendix A.4. After calculating the target distribution for all time steps, the critic can be trained to reduce the quantile regression loss with the target distribution. To provide more insight, we experiment with a toy example in Appendix A.6, and the results show that the proposed target distribution can trade off bias and variance through the trace-decay $\lambda$.

### 3.3 Safe Distributional Actor-Critic

Finally, we describe the proposed method, safe distributional actor-critic (SDAC). After collecting trajectories, the policy is updated by solving (5), which can be solved through a line search (for more detail, see Appendix A.8). The cost value and the cost square value functions in (4) can be obtained using the distributional critics as follows:

$$Q_C^\pi(s,a) = \int_{-\infty}^\infty z \Pr(Z_C^\pi(s,a) = z) dz \approx \frac{1}{M} \sum_{m=1}^M \theta_m(s,a),$$
$$S_C^\pi(s,a) = \int_{-\infty}^\infty z^2 \Pr(Z_C^\pi(s,a) = z) dz \approx \frac{1}{M} \sum_{m=1}^M \theta_m(s,a)^2. \quad (14)$$

If a solution of (5) does not exist, the policy is projected into a feasible region through the proposed gradient integration method. The critics can also be updated by the regression loss (7) between the target distribution obtained from (12). The proposed method is summarized in Algorithm 1.

## 4 Related Work

**Safe Reinforcement Learning.** García and Fernández [2015] and Gu et al. [2022] have researched and categorized safe RL methodologies from various perspectives. In this paper, we introduce safe RL

---

**Algorithm 1** Safe Distributional Actor-Critic

---

**Input:** Policy network $\pi_\psi$, reward and cost critic networks $Z_{R,\theta}^\pi$, $Z_{C_k,\theta}^\pi$, and replay buffer $\mathcal{D}$.
Initialize network parameters $\psi, \theta$, and replay buffer $\mathcal{D}$.
**for** epochs $= 1$ **to** $E$ **do**
   **for** $t = 1$ **to** $T$ **do**
      Sample $a_t \sim \pi_\psi(\cdot|s_t)$ and get $s_{t+1}, r_t = R(s_t, a_t, s_{t+1})$, and $c_{k,t} = C_k(s_t, a_t, s_{t+1}) \; \forall k$.
      Store $(s_t, a_t, \pi_\psi(a_t|s_t), r_t, c_{\{1,...,K\},t}, s_{t+1})$ in $\mathcal{D}$.
   **end for**
   Calculate the TD($\lambda$) target distribution (Section 3.2) using $\mathcal{D}$ and update the critics to minimize the quantile loss defined in (7).
   Calculate the surrogates for the objective and constraints defined in (37) using $\mathcal{D}$.
   Update the policy by solving (5), but if (5) has no solution, take a recovery step (Section 3.1).
**end for**

---

methods depending on how to update policies to reflect safety constraints. First, trust region-based safe RL methods [Achiam et al., 2017, Yang et al., 2020, Kim and Oh, 2022a] find policy update directions by approximating the safe RL problem as a linear-quadratic constrained linear program and update policies through a line search. Yang et al. [2020] also employ projection to meet a constraint; however, their method is limited to a single constraint and does not show to satisfy the constraint for the infeasible starting case. Second, Lagrangian-based methods [Stooke et al., 2020, Yang et al., 2021, Liu et al., 2020] convert the safe RL problem to a dual problem and update the policy and dual variables simultaneously. Last, expectation-maximization (EM) based methods [Liu et al., 2022, Zhang et al., 2022] find non-parametric policy distributions by solving the safe RL problem in E-steps and fit parametric policies to the found non-parametric distributions in M-steps. Also, there are other ways to reflect safety other than policy updates. Qin et al. [2021], Lee et al. [2022] find optimal state or state-action distributions that satisfy constraints, and Bharadhwaj et al. [2021], Thananjeyan et al. [2021] reflect safety during exploration by executing only safe action candidates. In the experiments, only the safe RL methods of the policy update approach are compared with the proposed method.

**Distributional TD($\lambda$).** TD($\lambda$) [Precup et al., 2000] can be extended to the distributional critic to trade off bias-variance. Gruslys et al. [2018] have proposed a method to obtain target distributions by mixing $n$-step distributions, but the method is applicable only in discrete action spaces. Nam et al. [2021] have proposed a method to obtain target distributions using sampling to apply to continuous action spaces, but this is only for on-policy settings. A method proposed by Tang et al. [2022] updates the critics using newly defined distributional TD errors rather than target distributions. This method is applicable for off-policy settings but has the disadvantage that memory usage increases linearly with the number of TD error steps. In contrast to these methods, the proposed method is memory-efficient and applicable for continuous action spaces under off-policy settings.

**Gradient Integration.** The proposed feasibility handling method utilizes a gradient integration method, which is widely used in multi-task learning (MTL). The gradient integration method finds a single gradient to improve all tasks by using gradients of all tasks. Yu et al. [2020] have proposed a projection-based gradient integration method, which is guaranteed to converge Pareto-stationary sets. A method proposed by Liu et al. [2021] can reflect user preference, and Navon et al. [2022] proposed a gradient-scale invariant method to prevent the training process from being biased by a few tasks. The proposed method can be viewed as a mixture of projection and scale-invariant methods as gradients are clipped and projected onto a trust region.

## 5 Experiments

We evaluate the safety performance of the proposed method in single- and multi-constrained robotic tasks. For single constraints, the agent performs four tasks provided by Safety Gym [Ray et al., 2019], and for multi-constraints, it performs bipedal and quadrupedal locomotion tasks.

### 5.1 Safety Gym

**Tasks.** We employ two robots, point and car, to perform goal and button tasks in the Safety Gym. The goal task is to control a robot toward a randomly spawned goal without passing through hazard

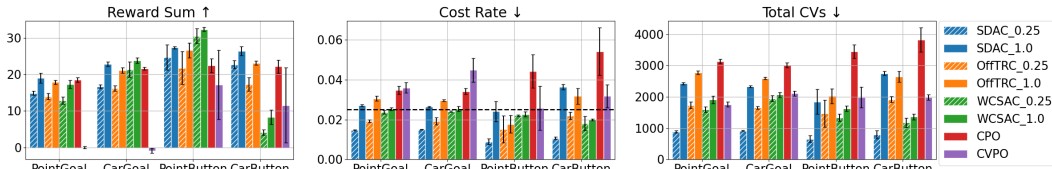

(a) Results of the final reward sum, cost rate, and total number of CVs. The number after the algorithm name in the legend indicates $\alpha$ used for the risk-averse constraint.

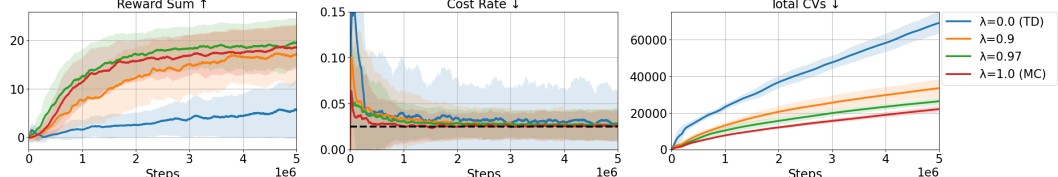

(b) Training curves of the point goal task according to the trace-decay $\lambda$. The solid line represents the average value, and the shaded area shows half of the std value.

Figure 3: **Safety Gym task results.** The cost rates show the cost sums divided by the episode length, and the dashed black lines indicate the threshold of the constraint. All methods are trained with five random seeds.

regions. The button task is to click a randomly designated button using a robot, where not only hazard regions but also dynamic obstacles exist. Agents get a cost when touching undesignated buttons and obstacles or entering hazard regions. There is only one constraint for the Safety Gym tasks, and it is defined using (4) with the sum of costs. Constraint violations (CVs) are counted when the cost sum exceeds the threshold. For more details, see Appendix B.

**Baselines.** Safe RL methods based on various types of policy updates are used as baselines. For the trust region-based method, we use constrained policy optimization (CPO) [Achiam et al., 2017] and off-policy trust-region CVaR (OffTRC) [Kim and Oh, 2022a], which extend the CPO to an off-policy and mean-std constrained version. For the Lagrangian-based method, distributional worst-case soft actor-critic (WCSAC) [Yang et al., 2022] is used, and constrained variational policy optimization (CVPO) [Liu et al., 2022] based on the EM method is used. Specifically, WCSAC, OffTRC, and the proposed method, SDAC, use the risk-averse constraints, so we experiment with those for $\alpha = 0.25$ and $1.0$ (when $\alpha = 1.0$, the constraint is identical to the risk-neutral constraint).

**Results.** The graph of the final reward sum, cost rate, and the total number of CVs are shown in Figure 3a, and the training curves are provided in Appendix C.1. We can interpret the results as good if the reward sum is high and the cost rate and total CVs are low. SDAC with $\alpha = 0.25$, risk-averse constraint situations, satisfies the constraints in all tasks and shows an average of 1.78 times fewer total CVs than the second-best algorithm. Nevertheless, since the reward sums are also in the middle or upper ranks, its safety performance is of high quality. SDAC with $\alpha = 1.0$, risk-neutral constraint situations, shows that the cost rates are almost the same as the thresholds except for the car button. In the case of the car button, the constraint is not satisfied, but by setting $\alpha = 0.25$, SDAC can achieve the lowest total CVs and the highest reward sum compared to the other methods. As for the reward sum, SDAC is the highest in the point goal and car button, and WCSAC is the highest in the rest. However, WCSAC seems to lose the risk-averse properties seeing that the cost rates do not change significantly according to $\alpha$. This is because WCSAC does not define constraints as risk measures of cost returns but as expectations of risk measures [Yang et al., 2022]. OffTRC has lower safety performance than SDAC in most cases because, unlike SDAC, it does not use distributional critics. Finally, CVPO and CPO are on-policy methods, so they are less efficient than the other methods.

### 5.2 Locomotion Tasks

**Tasks.** The locomotion tasks are to train robots to follow $xy$-directional linear and $z$-directional angular velocity commands. Mini-Cheetah from MIT [Katz et al., 2019] and Laikago from Unitree [Wang, 2018] are used for quadrupedal robots, and Cassie from Agility Robotics [Xie et al., 2018] is used for a bipedal robot. In order to perform the locomotion tasks, robots should keep balancing, standing, and stamping their feet so that they can move in any direction. Therefore, we define three constraints. The first is to keep the balance so that the body angle does not deviate from zero, and the second is to keep the height of CoM above a threshold. The third is to match the current foot

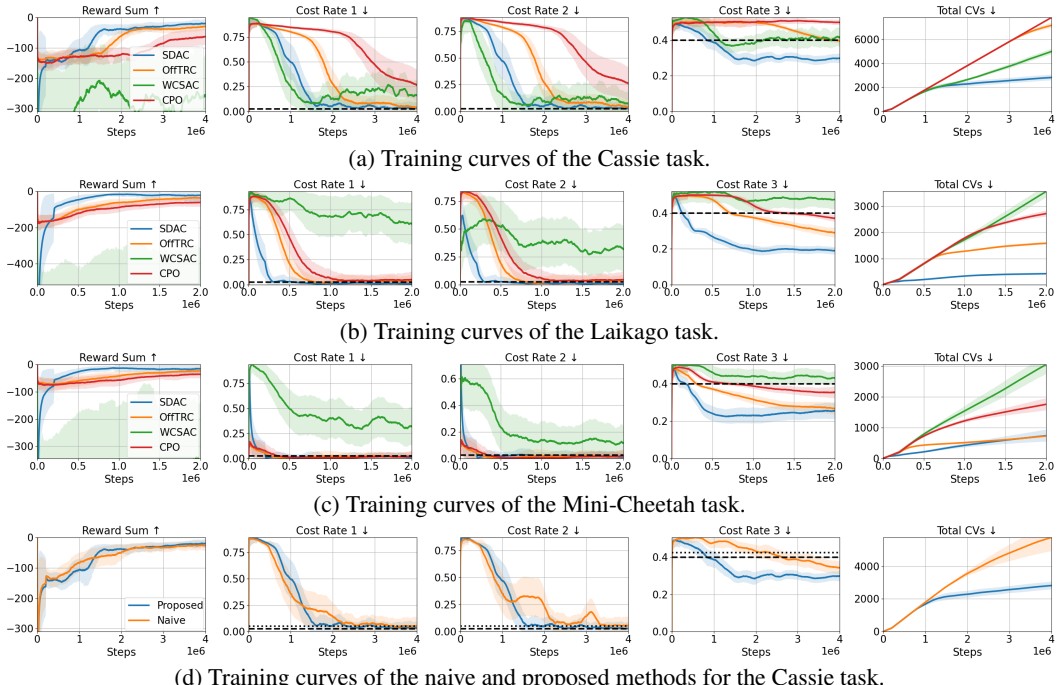

(a) Training curves of the Cassie task.

(b) Training curves of the Laikago task.

(c) Training curves of the Mini-Cheetah task.

(d) Training curves of the naive and proposed methods for the Cassie task.

Figure 4: **Locomotion task results.** The black dashed lines indicate the thresholds, and the dotted lines in (d) represent the thresholds $+ 0.025$. The shaded area represents half of the standard deviation, and all methods are trained with five random seeds.

contact state with a predefined foot contact timing. The reward is defined as the negative $l^2$-norm of the difference between the command and the current velocity. CVs are counted when the sum of at least one cost rate exceeds the threshold. For more details, see Appendix B.

**Baselines.** The baseline methods are identical to the Safety Gym tasks, and CVPO is excluded because it is technically challenging to scale to multiple constraint settings. We set $\alpha$ to 1 for the risk-averse constrained methods (OffTRC, WCSAC, and SDAC) to focus on measuring multi-constraint handling performance.

**Results.** Figure 4.(a-c) presents the training curves. SDAC shows the highest reward sums and the lowest total CVs in all tasks. In particular, the number of steps required to satisfy all constraints is 1.93 times fewer than the second-best algorithm on average. Trust region methods (OffTRC, CPO) stably satisfy constraints, but they are not efficient since they handle constraints by the naive approach. WCSAC, a Lagrangian method, fails to keep the constraints and shows the lowest reward sums. This is because the Lagrange multipliers can hinder the training stability due to the concurrent update with policy [Stooke et al., 2020].

## 5.3 Ablation Study

We conduct ablation studies to show whether the proposed target distribution lowers the estimation bias and whether the proposed gradient integration quickly converges to the feasibility condition. In Figure 3b, the number of CVs is reduced as $\lambda$ increases, which means that the bias of constraint estimation decreases. However, the score also decreases due to large variance, showing that $\lambda$ can adjust the bias-variance tradeoff. In Figure 4d, the proposed gradient integration method is compared with the naive approach, which minimizes the constraints in order from the first to the third constraint, as described in Section 3.1. The proposed method reaches the feasibility condition faster than the naive approach and shows stable training curves because it reflects all constraints concurrently. Additionally, we analyze the distributional critics in Appendix C.2 and the hyperparameters, such as the trust region size, in Appendix C.3. Furthermore, we analyze the sensitivity of traditional RL algorithms to reward configuration in Appendix D, emphasizing the advantage of safe RL that does not require reward tuning.

## 6  Limitation

A limitation of the proposed method is that the computational complexity of the gradient integration is proportional to the square of the number of constraints, whose qualitative analysis is presented in Appendix E.1. Also, we conducted quantitative analyses in Appendix E.2 by measuring wall clock training time. In the mini-cheetah task which has three constraints, the training time of SDAC is the third fastest among the four safe RL algorithms. Gradient integration is not applied when the policy satisfies constraints, so it may not constitute a significant proportion of training time. However, its influence can be dominant as the number of constraints increases. In order to resolve this limitation, the calculation can be speeded up by stochastically selecting a subset of constraints [Liu et al., 2021], or by reducing the frequency of policy updates [Navon et al., 2022]. The other limitation is that the mean-std defined in (2) is not a coherent risk measure. As a result, mean-std constraints can be served as reducing uncertainty rather than risk, although we experimentally showed that constraint violations are efficiently reduced. To resolve this, we can use the CVaR constraint, which can be estimated using an auxiliary variable, as done by Chow et al. [2017]. However, this solution can destabilize the training process due to the auxiliary variable, as observed in experiments of Kim and Oh [2022b]. Hence, a stabilization technique should be developed to employ the CVaR constraint.

## 7  Conclusion

We have presented the trust region-based safe distributional RL method, called *SDAC*. Through the locomotion tasks, it is verified that the proposed method efficiently satisfies multiple constraints using the gradient integration. Moreover, constraints can be stably satisfied in various tasks due to the low-biased distributional critics trained using the proposed target distributions. In addition, the proposed method is analyzed from multiple perspectives through various ablation studies. However, to compensate for the computational complexity, future work plans to devise efficient methods when dealing with large numbers of constraints.

## Acknowledgments and Disclosure of Funding

This work was partly supported by Institute of Information & Communications Technology Planning & Evaluation (IITP) grant funded by the Korea government (MSIT) (No. 2019-0-01190, [SW Star Lab] Robot Learning: Efficient, Safe, and Socially-Acceptable Machine Learning, 34%), Basic Science Research Program through the National Research Foundation of Korea (NRF) funded by the Ministry of Science and ICT (NRF-2022R1A2C2008239, General-Purpose Deep Reinforcement Learning Using Metaverse for Real World Applications, 33%), and Institute of Information & Communications Technology Planning & Evaluation (IITP) grant funded by the Korea government (MSIT) (No. 2021-0-01341, AI Graduate School Program, CAU, 33%).

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

# A  Algorithm Details

## A.1  Proof of Theorem 3.1

We denote the policy parameter space as $\Psi \subseteq \mathbb{R}^d$, the parameter at the $t$th iteration as $\psi_t \in \Psi$, the Hessian matrix as $H(\psi_t) = \nabla^2_\psi D_{\mathrm{KL}}(\pi_{\psi_t}||\pi_\psi)|_{\psi=\psi_t}$, and the $k$th constraint as $F_k(\psi_t) = F_k(\pi_{\psi_t}; \alpha)$. As we focus on the $t$th iteration, the following notations are used for brevity: $H = H(\psi_t)$ and $g_k = \nabla F_k(\psi_t)$. The proposed gradient integration at $t$th iteration is defined as the following quadratic program (QP):

$$g_t = \underset{g}{\mathrm{argmin}} \; \frac{1}{2} g^T H g \quad \text{s.t. } g_k^T g + c_k \leq 0 \text{ for } \forall k, \tag{15}$$

where $c_k = \min(\sqrt{2\epsilon g_k^T H^{-1} g_k}, F_k(\pi_\psi) - d_k + \zeta)$. In the remainder of this section, we introduce the assumptions and new definitions, discuss the existence of a solution (15), show the convergence to the feasibility condition for varying step size cases, and provide the proof of Theorem 3.1.

**Assumption.** **1)** Each $F_k$ is differentiable and convex, **2)** $\nabla F_k$ is $L$-Lipschitz continuous, **3)** all eigenvalues of the Hessian matrix $H(\psi)$ are equal or greater than $R \in \mathbb{R}_{>0}$ for $\forall \psi \in \Psi$, and **4)** $\{\psi | F_k(\psi) + \zeta < d_k \text{ for } \forall k\} \neq \emptyset$.

**Definition.** Using the Cholesky decomposition, the Hessian matrix can be expressed as $H = B \cdot B^T$ where $B$ is a lower triangular matrix. By introducing new terms, $\bar{g}_k := B^{-1} g_k$ and $b_t := B^T g_t$, the following is satisfied: $g_k^T H^{-1} g_k = ||\bar{g}_k||_2^2$. Additionally, we define the in-boundary and out-boundary sets as:

$$\mathrm{IB}_k := \left\{ \psi | F_k(\psi) - d_k + \zeta \leq \sqrt{2\epsilon \nabla F_k(\psi)^T H^{-1}(\psi) \nabla F_k(\psi)} \right\},$$

$$\mathrm{OB}_k := \left\{ \psi | F_k(\psi) - d_k + \zeta \geq \sqrt{2\epsilon \nabla F_k(\psi)^T H^{-1}(\psi) \nabla F_k(\psi)} \right\}.$$

The minimum of $||\bar{g}_k||$ in $\mathrm{OB}_k$ is denoted as $m_k$, and the maximum of $||\bar{g}_k||$ in $\mathrm{IB}_k$ is denoted as $M_k$. Also, $\min_k m_k$ and $\max_k M_k$ are denoted as $m$ and $M$, respectively, and we can say that $m$ is positive.

**Lemma A.1.** *For all $k$, the minimum value of $m_k$ is positive.*

*Proof.* Assume that there exist $k \in \{1, ..., K\}$ such that $m_k$ is equal to zero at a policy parameter $\psi^* \in \mathrm{OB}_k$, i.e., $||\nabla F_k(\psi^*)|| = 0$. Since $F_k$ is convex, $\psi^*$ is a minimum point of $F_k$, $\min_\psi F_k(\psi) = F_k(\psi^*) < d_k - \zeta$. However, $F_k(\psi^*) \geq d_k - \zeta$ as $\psi^* \in \mathrm{OB}_k$, so $m_k$ is positive due to the contradiction. Hence, the minimum of $m_k$ is also positive. $\qquad\square$

**Lemma A.2.** *A solution of (15) always exists.*

*Proof.* There exists a policy parameter $\hat{\psi} \in \{\psi | F_k(\psi) + \zeta < d_k \text{ for } \forall k\}$ due to the assumptions. Let $g = \psi - \psi_t$. Then, the following inequality holds.

$$g_k^T(\psi - \psi_t) + c_k \leq g_k^T(\psi - \psi_t) + F_k(\psi_t) + \zeta - d_k \leq F_k(\psi) + \zeta - d_k. \quad (\because F_k \text{ is convex.})$$
$$\Rightarrow g_k^T(\hat{\psi} - \psi_t) + c_k \leq F_k(\hat{\psi}) + \zeta - d_k < 0 \text{ for } \forall k.$$

Since $\hat{\psi} - \psi_t$ satisfies all constraints of (15), the feasible set is non-empty and convex. Also, $H$ is positive definite, so the QP has a unique solution. $\qquad\square$

Lemma A.2 shows the existence of solution of (15). Now, we show the convergence of the proposed gradient integration method in the case of varying step sizes.

**Lemma A.3.** *If $\sqrt{2\epsilon}M \leq \zeta$ and a policy is updated by $\psi_{t+1} = \psi_t + \beta_t g_t$, where $0 < \beta_t < \frac{2\sqrt{2\epsilon}mR}{L||b_t||^2}$ and $\beta_t \leq 1$, the policy satisfies $F_k(\psi) \leq d_k$ for $\forall k$ within a finite time.*

*Proof.* We can reformulate the step size as $\beta = \frac{2\sqrt{2\epsilon}mR}{L||b_t||^2}\beta'_t$, where $\beta'_t \leq \frac{L||b_t||^2}{2\sqrt{2\epsilon}mR}$ and $0 < \beta'_t < 1$. Since the eigenvalues of $H$ is equal to or bigger than $R$ and $H$ is symmetric and positive definite,

$\frac{1}{R}I - H^{-1}$ is positive semi-definite. Hence, $x^T H^{-1} x \leq \frac{1}{R}||x||^2$ is satisfied. Using this fact, the following inequality holds:

$$F_k(\psi_t + \beta_t g_t) - F_k(\psi_t) \leq \beta_t \nabla F_k(\psi_t)^T g_t + \frac{L}{2}||\beta_t g_t||^2 \quad (\because \nabla F_k \text{ is } L\text{-Lipschitz continuous.})$$

$$= \beta_t g_k^T g_t + \frac{L}{2}\beta_t^2||g_t||^2$$

$$= \beta_t g_k^T g_t + \frac{L}{2}\beta_t^2 b_t^T H^{-1} b_t \quad (\because g_t = B^{-T} b_t)$$

$$\leq -\beta_t c_k + \frac{L}{2R}\beta_t^2||b_t||^2. \quad (\because g_k^T g_t + c_k \leq 0)$$

Now, we will show that $\psi$ enters $\text{IB}_k$ in a finite time for $\forall \psi \in \text{OB}_k$ and that the $k$th constraint is satisfied for $\forall \psi \in \text{IB}_k$. Thus, we divide into two cases, **1)** $\psi_t \in \text{OB}_k$ and **2)** $\psi_t \in \text{IB}_k$. For the first case, $c_k = \sqrt{2\epsilon}||\bar{g}_k||$, so the following inequality holds:

$$F_k(\psi_t + \beta_t g_t) - F_k(\psi_t) \leq \beta_t \left( -\sqrt{2\epsilon}||\bar{g}_k|| + \frac{L}{2R}\beta_t||b_t||^2 \right)$$

$$\leq \beta_t \sqrt{2\epsilon} \left( -||\bar{g}_k|| + m\beta_t' \right) \tag{16}$$

$$\leq \beta_t \sqrt{2\epsilon} m(\beta_t' - 1) < 0.$$

The value of $F_k$ decreases strictly with each update step according to (16). Hence, $\psi_t$ can reach $\text{IB}_k$ by repeatedly updating the policy. We now check whether the constraint is satisfied for the second case. For the second case, the following inequality holds by applying $c_k = F_k(\psi_t) - d_k + \zeta$:

$$F_k(\psi_t + \beta_t g_t) - F_k(\psi_t) \leq \beta_t d_k - \beta_t F_k(\psi_t) - \beta_t \zeta + \frac{L}{2R}\beta_t^2||b_t||^2$$

$$\Rightarrow F_k(\psi_t + \beta_t g_t) - d_k \leq (1 - \beta_t)(F_k(\psi_t) - d_k) + \beta_t(-\zeta + \sqrt{2\epsilon}m\beta_t').$$

Since $\psi_t \in \text{IB}_k$,

$$F_k(\psi_t) - d_k \leq \sqrt{2\epsilon}||\bar{g}_k|| - \zeta \leq \sqrt{2\epsilon}M - \zeta \leq 0.$$

Since $m \leq M$ and $\beta_t' < 1$,

$$-\zeta + \sqrt{2\epsilon}m\beta_t' < -\zeta + \sqrt{2\epsilon}M \leq 0.$$

Hence, $F_k(\psi_t + \beta_t g_t) \leq d_k$, which means that the $k$th constraint is satisfied if $\psi_t \in \text{IB}_k$. As $\psi_t$ reaches $\text{IB}_k$ for $\forall k$ within a finite time according to (16), the policy can satisfy all constraints within a finite time. $\qquad \square$

Lemma A.3 shows the convergence to the feasibility condition in the case of varying step sizes. We introduce a lemma, which shows $||b_t||$ is bounded by $\sqrt{\epsilon}$, and finally show the proof of Theorem 3.1, which can be considered a special case of varying step sizes.

**Lemma A.4.** *There exists $T \in \mathbb{R}_{>0}$ such that $||b_t|| \leq T\sqrt{\epsilon}$.*

*Proof.* Let us define the following sets:

$$I := \{k | F_k(\psi_t) + \zeta - d_k < 0\}, \ O := \{k | F_k(\psi_t) + \zeta - d_k > 0\},$$

$$U := \{k | F_k(\psi_t) + \zeta - d_k = 0\}, \ C(\epsilon) := \{g | g_k^T g + c_k(\epsilon) \leq 0 \ \forall k\}, \tag{17}$$

$$I_G := \{g | g_k^T g + F_k(\psi_t) + \zeta - d_k \leq 0 \ \forall k \in I\},$$

where $c_k(\epsilon) = \min(\sqrt{2\epsilon g_k^T H^{-1} g_k}, F_k(\pi_\psi) - d_k + \zeta)$. Using these sets, the following vectors can be defined: $g(\epsilon) := \text{argmin}_{g \in C(\epsilon)} \frac{1}{2}g^T H g$, $b(\epsilon) := B^T g(\epsilon)$. Now, we will show that $||b(\epsilon)||$ is bounded above and $||b(\epsilon)|| \propto \sqrt{\epsilon}$ for sufficiently small $\epsilon > 0$.

First, the following is satisfied for a sufficiently large $\epsilon$:

$$C(\epsilon) = \{g | g_k^T g + F_k(\psi_t) + \zeta - d_k \leq 0 \ \forall k\}. \tag{18}$$

Since $\hat{\psi} - \psi_t \in C(\epsilon)$, where $\hat{\psi}$ is defined in Lemma A.2, $||b(\epsilon)|| \leq \frac{1}{2}(\hat{\psi} - \psi_t)^T H(\hat{\psi} - \psi_t)$ for $\forall \epsilon$. Therefore, $||b(\epsilon)||$ is bounded above.

Second, let us define the following trust region size:

$$\hat{\epsilon} := \frac{1}{2}\left(\min_{k \in O} \frac{F_k(\psi_t) + \zeta - d_k}{||\bar{g}_k||}\right)^2 > 0. \tag{19}$$

if , $\epsilon \leq \hat{\epsilon}$, the following is satisfied:

$$C(\epsilon) = I_G \cap \{g|g_k^T g + \sqrt{2\epsilon}||\bar{g}_k|| \leq 0 \ \forall k \in O, \ g_k^T g \leq 0 \ \forall k \in U\} \neq \phi. \tag{20}$$

Thus, $O_G(\epsilon) := \{g|g_k^T g + \sqrt{2\epsilon}||\bar{g}_k|| \leq 0 \ \forall k \in O, \ g_k^T g \leq 0 \ \forall k \in U\}$ is not empty. If we define $\hat{g}(\epsilon) := \text{argmin}_{g \in O_G(\epsilon)} \frac{1}{2}g^T H g$, the following is satisfied:

$$\hat{g}(\epsilon) = \sqrt{\frac{\epsilon}{\hat{\epsilon}}}\hat{g}(\hat{\epsilon}) \text{ for } 0 \leq \epsilon \leq \hat{\epsilon}. \tag{21}$$

Then, if $\epsilon \leq \hat{\epsilon}\left(\min_{k \in I}(d_k - \zeta - F_k(\psi_t))/g_k^T \hat{g}(\hat{\epsilon}))\right)^2$, $\hat{g}(\epsilon) = g(\epsilon)$ since $\hat{g}(\epsilon) \in I_G$. Consequently, by defining a trust region size:

$$\epsilon^* := \hat{\epsilon} \cdot \min\left(1, \left(\min_{k \in I} \frac{d_k - \zeta - F_k(\psi_t)}{g_k^T \hat{g}(\hat{\epsilon})}\right)^2\right) > 0, \tag{22}$$

$g(\epsilon) = \sqrt{\epsilon/\hat{\epsilon}}\hat{g}(\hat{\epsilon})$ for $\epsilon \leq \epsilon^*$. Therefore, $||b(\epsilon)|| \propto \sqrt{\epsilon}$ if $\epsilon \leq \epsilon^*$.

Finally, since $||b(\epsilon)||$ is bounded above and proportional to $\sqrt{\epsilon}$ for sufficiently small $\epsilon$, there exist a constant $T$ such that $||b(\epsilon)|| \leq T\sqrt{\epsilon}$. $\qquad \square$

**Theorem 3.1.** *Assume that the constraints are differentiable and convex, gradients of the constraints are L-Lipschitz continuous, eigenvalues of the Hessian are equal or greater than a positive value $R \in \mathbb{R}_{>0}$, and $\{\psi|F_k(\pi_\psi; \alpha) + \zeta < d_k, \ \forall k\} \neq \emptyset$. Then, there exists $E \in \mathbb{R}_{>0}$ such that if $0 < \epsilon \leq E$ and a policy is updated by the proposed gradient integration method, all constraints are satisfied within finite time steps.*

*Proof.* The proposed step size is $\beta_t = \min(1, \sqrt{2\epsilon}/||b_t||)$, and the sufficient conditions that guarantee the convergence according to Lemma A.3 are followings:

$$\sqrt{2\epsilon}M \leq \zeta \text{ and } 0 < \beta_t \leq 1 \text{ and } \beta_t < \frac{2\sqrt{2\epsilon}mR}{L||b_t||^2}.$$

The second condition is self-evident. To satisfy the third condition, the proposed step size $\beta_t$ should satisfy the followings:

$$\frac{\sqrt{2\epsilon}}{||b_t||} < \frac{2\sqrt{2\epsilon}mR}{L||b_t||^2} \ \Leftrightarrow \ ||b_t|| < \frac{2mR}{L}.$$

If $\epsilon < 4((mR)/(LT))^2$, the following inequality holds:

$$\sqrt{\epsilon} < \frac{2mR}{LT} \ \Rightarrow \ ||b_t|| \leq T\sqrt{\epsilon} < \frac{2mR}{L}. \quad (\because \text{ Lemma A.4.})$$

Hence, if $\epsilon \leq E = \frac{1}{2}\min(\frac{\zeta^2}{2M^2}, 4(\frac{mR}{LT})^2)$, the sufficient conditions are satisfied. $\qquad \square$

## A.2 Toy Example for Gradient Integration Method

The problem of the toy example in Figure 1 is defined as:

$$\underset{x_1, x_2}{\text{minimize}} \sqrt{(\sqrt{3}x_1 + x_2 + 2)^2 + 4(x_1 - \sqrt{3}x_2 + 4)^2} \quad \textbf{s.t.} \ x_1 \geq 0, \ x_1 - 2x_2 \leq 0, \tag{23}$$

where there are two linear constraints. The initial points for the naive and gradient integration methods are $x_1 = -2.5$ and $x_2 = -3.0$, which do not satisfied the two constraints. We use the Hessian matrix for the trust region as identity matrix and the trust region size as $0.5$ in both methods. The naive method minimizes the constraints in order from the first to the second constraint.

### A.3 Analysis of Worst-Case Time to Satisfy All Constraints

To analyze the sample complexity, we consider a tabular MDP and use softmax policy parameterization as follows (for more details, see [Xu et al., 2021]):

$$\pi_\psi(a|s) := \frac{\exp \psi(s, a)}{\sum_{a'} \exp \psi(s, a')} \ \forall (s, a) \in S \times A. \tag{24}$$

According to Agarwal et al. [2021], the natural policy gradient (NPG) update is as follows:

$$\psi_{t+1} = \psi_t + \beta A^{\pi_{\psi_t}}, \ \pi_{\psi_{t+1}}(a|s) = \pi_{\psi_t}(a|s) \frac{\exp(\beta A^{\pi_{\psi_t}}(s, a))}{Z_t(s)}, \tag{25}$$

where $\beta$ is a step size, $A^{\pi_{\psi_t}} \in \mathbb{R}^{|S||A|}$ is the vector expression of the advantage function, and $Z_t(s) = \sum_a \pi_{\psi_t}(a|s) \exp(\beta A^{\pi_{\psi_t}}(s, a))/(1 - \gamma)$. Analyzing the sample complexity of trust region-based methods is challenging since their stepsize is not fixed, so we modify the gradient integration method to use the NPG as follows:

$$g^* = \operatorname{argmin}_g \frac{1}{2} g^T H g \text{ s.t. } g_k^T g + c_k \leq 0 \ \forall k \in \{k | F_k(\pi_{\psi_t}; \alpha) > d_k\},$$
$$\psi_{t+1} = \psi_t + \beta g^*/||g^*||_2. \tag{26}$$

In the remainder, we abbreviate $\pi_{\psi_t}$, $A^{\pi_{\psi_t}}$, and $F_k(\pi_{\psi_t}; \alpha)$ as $\pi_t$, $A^t$, and $F_k(\pi_t)$, respectively. Since $g^*$ always exists due to Lemma A.2, we can write the policy using Lagrange multipliers $\lambda_k^t \geq 0$ as follows:

$$\psi_{t+1} = \psi_t - \beta \sum_k \lambda_k^t A_{C_k}^t / W_t, \ W_t := ||\sum_k \lambda_k^t A_{C_k}^t||_2,$$
$$\pi_{t+1}(a|s) = \pi_t(a|s) \exp\left(-\frac{\beta}{W_t} \sum_k \lambda_k^t A_{C_k}^t(s, a)\right)/Z_t(s), \tag{27}$$

where $Z_t(s)$ is a normalization factor, and $\lambda_k^t = 0$ for $F_k(\pi_t) \leq d_k$. The naive approach can also be written as above, except that $\lambda^t$ is a one-hot vector, where $i$-th value $\lambda_i^t$ is one only for corresponding to the randomly selected constraint. Then, we can get the followings:

$$\sum_k \lambda_k^t (F_k(\pi_{t+1}) - F_k(\pi_t))/W_t = \frac{1}{1 - \gamma} \mathbb{E}_{s \sim d^{\pi_{t+1}}} \left[\sum_a \pi_{t+1}(a|s) \sum_k \lambda_k^t A_{C_k}^t(s, a)/W_t\right]$$

$$= -\frac{1}{\beta(1 - \gamma)} \mathbb{E}_{s \sim d^{\pi_{t+1}}} \left[\sum_a \pi_{t+1}(a|s) \log \frac{\pi_{t+1}(a|s) Z_t(s)}{\pi_t(a|s)}\right]$$

$$= -\frac{1}{\beta(1 - \gamma)} \mathbb{E}_{s \sim d^{\pi_{t+1}}} \left[D_{\mathrm{KL}}(\pi_{t+1}(\cdot|s)||\pi_t(\cdot|s)) + \log Z_t(s)\right] \tag{28}$$

$$\leq -\frac{1}{\beta(1 - \gamma)} \mathbb{E}_{s \sim d^{\pi_{t+1}}} \left[\log Z_t(s)\right] \qquad (\because D_{\mathrm{KL}}(\pi'||\pi) \geq 0)$$

$$\leq -\frac{1}{\beta} \mathbb{E}_{s \sim \rho} \left[\log Z_t(s)\right], \qquad (\because ||d^\pi/\rho||_\infty \geq 1 - \gamma)$$

We can also get the followings by using the Lemma 7 in Xu et al. [2021]:

$$\sum_k \lambda_k^t (F_k(\pi_*) - F_k(\pi_t))/W_t \geq -\frac{1}{\beta(1 - \gamma)} \mathbb{E}_{s \sim d^*} \left[D_{\mathrm{KL}}(\pi_*||\pi_t) - D_{\mathrm{KL}}(\pi_*||\pi_{t+1})\right]$$
$$- \sum_k \lambda_k^t \frac{2\beta C_{\max}}{(1 - \gamma)^2 W_t}, \tag{29}$$

where $\pi_*$ is an optimal policy, and $C_{\max}$ is the maximum value of costs. If $\lambda_k^t > 0$, $F_k(\pi_t) - F_k(\pi_*) > \zeta$. Thus, $\sum_k \lambda_k^t (F_k(\pi_*) - F_k(\pi_t)) \leq -\zeta \sum_k \lambda_k^t$. If the policy does not satisfy the constraints until $T$ step, the following inequality holds by summing the above inequalities from $t = 0$ to $T$:

$$\beta(1 - \gamma)(\zeta - \frac{2\beta C_{\max}}{(1 - \gamma)^2}) \sum_{t=0}^{T} \sum_i \lambda_i^t / W_t \leq \mathbb{E}_{s \sim d^*} \left[D_{\mathrm{KL}}(\pi_*||\pi_0)\right]. \tag{30}$$

Let denote $\frac{1}{T}\sum_{t=0}^{T}\sum_i \lambda_i^t/W_t$ as $\mathbb{E}_t[\sum_i \lambda_i^t/W_t]$, and we can get $W_t = ||\sum_k \lambda_k^t A_{C_k}^t||_2 \leq \sum_k \lambda_k^t 2|S||A|C_{\max}/(1-\gamma)$. Then, the maximum $T$ can be expressed as:

$$T \leq \frac{D_{\mathrm{KL}}}{\beta(1-\gamma)(\zeta - \frac{2\beta C_{\max}}{(1-\gamma)^2})\mathbb{E}_t[\sum_i \lambda_i^t/W_t]} \leq \frac{2|S||A|C_{\max}D_{\mathrm{KL}}}{\beta(1-\gamma)^2\zeta - 2\beta^2 C_{\max}} =: T_{\max}, \qquad (31)$$

where we abbreviate $\mathbb{E}_{s\sim d^*}[D_{\mathrm{KL}}(\pi_*||\pi_0)]$ as $D_{\mathrm{KL}}$. Finally, the policy can reach the feasible region within $T_{\max}$ steps.

The worst-case time of the naive approach is the same as the above equation, except for the $\lambda$ part. In the naive approach, $\lambda^t$ is a one-hot vector, as mentioned earlier. In other words, only $\mathbb{E}_t[\sum_i \lambda_i^t/W_t] = \mathbb{E}_t[\sum_i \lambda_i^t/||\sum_i \lambda_i^t A_{C_k}^t||_2]$ is different. Let us assume that the advantage vector follows a normal distribution. Then, the variance of $\sum_i \lambda_i^t A_{C_k}^t$ is smaller for $\lambda^t$ with distributed values than for one-hot values. Then, the reciprocal of the 2-norm becomes larger, resulting in a decrease in the worst-case time. From this perspective, the gradient integration method has a benefit over the naive approach as it reduces the variance of the advantage vector. Even though we cannot officially say that the worst-case time of the proposed method is smaller than the naive method because the advantage vector does not follow the normal distribution, we can deliver our insight on the benefit of gradient integration method.

### A.4   Proof of Theorem 3.2

In this section, we show that a sequence, $Z_{k+1} = \mathcal{T}_\lambda^{\mu,\pi} Z_k$, converges to the $Z_R^\pi$. First, we rewrite the operator $\mathcal{T}_\lambda^{\mu,\pi}$ for random variables to an operator for distributions and show that the operator is contractive. Finally, we show that $Z_R^\pi$ is the unique fixed point.

Before starting the proof, we introduce useful notions and distance metrics. As the return $Z_R^\pi(s,a)$ is a random variable, we define the distribution of $Z_R^\pi(s,a)$ as $\nu_R^\pi(s,a)$. Let $\eta$ be the distribution of a random variable $X$. Then, we can express the distribution of affine transformation of random variable, $aX + b$, using the *pushforward* operator, which is defined by Rowland et al. [2018], as $(f_{a,b})_{\#}(\eta)$. To measure a distance between two distributions, Bellemare et al. [2023] has defined the distance $l_p$ as follows:

$$l_p(\eta_1, \eta_2) := \left( \int_{\mathbb{R}} |F_{\eta_1}(x) - F_{\eta_2}(x)|^p \, dx \right)^{1/p}, \qquad (32)$$

where $F_\eta(x)$ is the cumulative distribution function. This distance is $1/p$-homogeneous, regular, and $p$-convex (see Section 4 of Bellemare et al. [2023] for more details). For functions that map state-action pairs to distributions, a distance can be defined as [Bellemare et al., 2023]: $\bar{l}_p(\nu_1, \nu_2) := \sup_{(s,a)\in S\times A} l_p(\nu_1(s,a), \nu_2(s,a))$. Then, we can rewrite the operator $\mathcal{T}_\lambda^{\mu,\pi}$ for random variables in (13) as an operator for distributions as below.

$$
\begin{aligned}
\mathcal{T}_\lambda^{\mu,\pi}\nu(s,a) &:= \frac{1-\lambda}{\mathcal{N}}\sum_{i=0}^\infty \lambda^i \\
&\times \mathbb{E}_\mu\left[ \left( \prod_{j=1}^i \eta(s_j, a_j) \right) \mathbb{E}_{a'\sim\pi(\cdot|s_{i+1})}\left[ (f_{\gamma^{i+1}, \sum_{t=0}^i \gamma^t r_t})_{\#}(\nu(s_{i+1}, a')) \right] \Big| s_0 = s, a_0 = a \right],
\end{aligned}
\qquad (33)
$$

where $\eta(s,a) = \frac{\pi(a|s)}{\mu(a|s)}$ and $\mathcal{N}$ is a normalization factor. Since the random variable $Z(s,a)$ and the distribution $\nu(s,a)$ is equivalent, the operators in (13) and (33) are also equivalent. Hence, we are going to show the proof of Theorem 3.2 using (33) instead of (13). We first show that the operator $\mathcal{T}_\lambda^{\mu,\pi}$ has a contraction property.

**Lemma A.5.** *Under the distance $\bar{l}_p$ and the assumption that the state, action, and reward spaces are finite, $\mathcal{T}_\lambda^{\mu,\pi}$ is $\gamma^{1/p}$-contractive.*

*Proof.* First, the operator can be rewritten using summation as follows.

$$
\mathcal{T}_\lambda^{\mu,\pi}\nu(s,a) = \frac{1-\lambda}{\mathcal{N}}\sum_{i=0}^\infty \lambda^i \sum_{a'\in A}\sum_{(s_0,a_0,r_0,...,s_{i+1})} \underbrace{\Pr{}_\mu(s_0,a_0,r_0,...,s_{i+1})}_{=:\tau}\left(\prod_{j=1}^i \eta(s_j,a_j)\right)
$$
$$
\times\, \pi(a'|s_{i+1})(f_{\gamma^{i+1},\sum_{t=0}^i \gamma^t r_t})\#(\nu(s_{i+1},a'))
$$
$$
= \frac{1-\lambda}{\mathcal{N}}\sum_{i=0}^\infty \lambda^i \sum_{a'\in A}\sum_\tau \Pr{}_\mu(\tau)\left(\prod_{j=1}^i \eta(s_j,a_j)\right)\pi(a'|s_{i+1})\sum_{s'\in S}\mathbf{1}_{s'=s_{i+1}}
$$
$$
\times \sum_{r'_{0:i}}\left(\prod_{k=0}^i \mathbf{1}_{r'_k=r_k}\right)(f_{\gamma^{i+1},\sum_{t=0}^i \gamma^t r'_t})\#(\nu(s',a'))
$$
$$\tag{34}$$
$$
= \frac{1-\lambda}{\mathcal{N}}\sum_{i=0}^\infty \lambda^i \sum_{a'\in A}\sum_{s'\in S}\sum_{r'_{0:i}}(f_{\gamma^{i+1},\sum_{t=0}^i \gamma^t r'_t})\#(\nu(s',a'))
$$
$$
\times\, \underbrace{\mathbb{E}_\mu\left[\left(\prod_{j=1}^i \eta(s_j,a_j)\right)\pi(a'|s_{i+1})\mathbf{1}_{s'=s_{i+1}}\left(\prod_{k=0}^i \mathbf{1}_{r'_k=r_k}\right)\right]}_{=:w_{s',a',r'_{0:i}}}
$$
$$
= \frac{1-\lambda}{\mathcal{N}}\sum_{i=0}^\infty \sum_{s'\in S}\sum_{a'\in A}\sum_{r'_{0:i}}\lambda^i w_{s',a',r'_{0:i}}(f_{\gamma^{i+1},\sum_{t=0}^i \gamma^t r'_t})\#(\nu(s',a')).
$$

Since the sum of weights of distributions should be one, we can find the normalization factor $\mathcal{N} = (1-\lambda)\sum_{i=0}^\infty\sum_{s\in S}\sum_{a\in A}\sum_{r_{0:i}}\lambda^i w_{s,a,r_{0:i}}$. Then, the following inequality can be derived using the homogeneity, regularity, and convexity of $l_p$:

$$
l_p^p(\mathcal{T}_\lambda^{\mu,\pi}\nu_1(s,a),\mathcal{T}_\lambda^{\mu,\pi}\nu_2(s,a))
$$
$$
= l_p^p\left(\frac{1-\lambda}{\mathcal{N}}\sum_{i=0}^\infty\sum_{s\in S}\sum_{a\in A}\sum_{r_{0:i}}\lambda^i w_{s,a,r_{0:i}}(f_{\gamma^{i+1},\sum_{t=0}^i \gamma^t r_t})\#(\nu_1(s,a)),\right.
$$
$$
\left.\frac{1-\lambda}{\mathcal{N}}\sum_{i=0}^\infty\sum_{s\in S}\sum_{a\in A}\sum_{r_{0:i}}\lambda^i w_{s,a,r_{0:i}}(f_{\gamma^{i+1},\sum_{t=0}^i \gamma^t r_t})\#(\nu_2(s,a))\right)
$$
$$
\le \sum_{i=0}^\infty\sum_{s\in S}\sum_{a\in A}\sum_{r_{0:i}}\frac{(1-\lambda)\lambda^i w_{s,a,r_{0:i}}}{\mathcal{N}}l_p^p\left((f_{\gamma^{i+1},\sum_{t=0}^i \gamma^t r_t})\#(\nu_1(s,a)),\right.
$$
$$
\left.(f_{\gamma^{i+1},\sum_{t=0}^i \gamma^t r_t})\#(\nu_2(s,a))\right)
$$
$$\tag{35}$$
$$
\le \sum_{i=0}^\infty\sum_{s\in S}\sum_{a\in A}\sum_{r_{0:i}}\frac{(1-\lambda)\lambda^i w_{s,a,r_{0:i}}}{\mathcal{N}}l_p^p\left((f_{\gamma^{i+1},0})\#(\nu_1(s,a)),(f_{\gamma^{i+1},0})\#(\nu_2(s,a))\right)
$$
$$
= \sum_{i=0}^\infty\sum_{s\in S}\sum_{a\in A}\sum_{r_{0:i}}\frac{(1-\lambda)\lambda^i w_{s,a,r_{0:i}}}{\mathcal{N}}\gamma^{i+1}l_p^p(\nu_1(s,a),\nu_2(s,a))
$$
$$
\le \sum_{i=0}^\infty\sum_{s\in S}\sum_{a\in A}\sum_{r_{0:i}}\frac{(1-\lambda)\lambda^i w_{s,a,r_{0:i}}}{\mathcal{N}}\gamma^{i+1}\left(\bar{l}_p(\nu_1,\nu_2)\right)^p
$$
$$
\le \gamma\left(\bar{l}_p(\nu_1,\nu_2)\right)^p.
$$

Therefore, $\bar{l}_p(\mathcal{T}_\lambda^{\mu,\pi}\nu_1,\mathcal{T}_\lambda^{\mu,\pi}\nu_2) \le \gamma^{1/p}\bar{l}_p(\nu_1,\nu_2)$. $\qquad\square$

By the Banach's fixed point theorem, the operator $\mathcal{T}_\lambda^{\mu,\pi}$ has a unique fixed distribution. We now show that the fixed distribution is $\nu_R^\pi$.

**Lemma A.6.** *The fixed distribution of the operator $\mathcal{T}_\lambda^{\mu,\pi}$ is $\nu_R^\pi$.*

*Proof.* From the definition of $Z_R^\pi$, the following equality holds [Rowland et al., 2018]: $\nu_R^\pi(s,a) = \mathbb{E}_\pi\left[(f_{\gamma,r})_{\#}(\nu_R^\pi(s',a'))\right]$. Then, it can be shown that $\nu_R^\pi$ is the fixed distribution by applying the operator $\mathcal{T}_\lambda^{\mu,\pi}$ to $\nu_R^\pi$:

$$\mathcal{T}_\lambda^{\mu,\pi}\nu_R^\pi(s,a) = \frac{1-\lambda}{\mathcal{N}}\sum_{i=0}^{\infty}\lambda^i$$

$$\times \mathbb{E}_\mu\left[\left(\prod_{j=1}^{i}\eta(s_j,a_j)\right)\mathbb{E}_{a'\sim\pi(\cdot|s_{i+1})}\left[(f_{\gamma^{i+1},\sum_{t=0}^{i}\gamma^t r_t})_{\#}(\nu_R^\pi(s_{i+1},a'))\right]\bigg| s_0=s, a_0=a\right]$$

$$= \frac{1-\lambda}{\mathcal{N}}\sum_{i=0}^{\infty}\lambda^i\mathbb{E}_\pi\left[(f_{\gamma^{i+1},\sum_{t=0}^{i}\gamma^t r_t})_{\#}(\nu_R^\pi(s_{i+1},a_{i+1}))\bigg| s_0=s, a_0=a\right]$$

$$= \frac{1-\lambda}{\mathcal{N}}\sum_{i=0}^{\infty}\lambda^i\nu_R^\pi(s,a) = \nu_R^\pi(s,a). \tag{36}$$

$\square$

**Theorem 3.2.** *Let define a distributional operator $\mathcal{T}_\lambda^{\mu,\pi}$, whose probability density function is:*

$$\Pr(\mathcal{T}_\lambda^{\mu,\pi}Z(s,a)=z) \propto$$
$$\sum_{i=0}^{\infty}\mathbb{E}_\mu\left[\lambda^i\prod_{j=1}^{i}\frac{\pi(a_j|s_j)}{\mu(a_j|s_j)}\mathbb{E}_{a'\sim\pi(\cdot|s_{i+1})}\left[\Pr\left(\sum_{t=0}^{i}\gamma^t R_t+\gamma^{i+1}Z(s_{i+1},a')=z\right)\right]\bigg| s_0=s,a_0=a\right]. \tag{13}$$

*Then, a sequence, $Z_{k+1}(s,a) = \mathcal{T}_\lambda^{\mu,\pi}Z_k(s,a) \ \forall(s,a)$, converges to $Z_R^\pi$.*

*Proof.* The operator $\mathcal{T}_\lambda^{\mu,\pi}$ is $\gamma^{1/p}$-contractive under the distance $\bar{l}_p$ according to Lemma A.5. Also, the fixed distribution of the operator is $\nu_R^\pi$, which is equivalent to $Z_R^\pi$, according to Lemma A.6. By the Banach's fixed point theorem, the sequence, $Z_{k+1}(s,a) = \mathcal{T}_\lambda^{\mu,\pi}Z_k(s,a) \ \forall(s,a)$, converges to the fixed distribution of the operator, $Z_R^\pi$. $\square$

### A.5 Pseudocode of TD($\lambda$) Target Distribution

We provide the pseudocode for calculating TD($\lambda$) target distribution for the reward critic in Algorithm 2. The target distribution for the cost critics can also be obtained by simply replacing the reward part with the cost.

---
**Algorithm 2** TD($\lambda$) Target Distribution

---

**Input:** Policy network $\pi_\psi$, critic network $Z_\theta^\pi$, and trajectory $\{(s_t,a_t,\mu(a_t|s_t),r_t,d_t,s_{t+1})\}_{t=1}^{T}$.
Sample an action $a'_{T+1}\sim\pi_\psi(s_{T+1})$ and get $\hat{Z}_T^{\text{tot}} = r_T + (1-d_T)\gamma Z_\theta^\pi(s_{T+1},a'_{T+1})$.
Initialize the total weight $w_{\text{tot}} = \lambda$.
**for** $t = T$ **to** 1 **do**
    Sample an action $a'_{t+1}\sim\pi_\psi(s_{t+1})$ and get $\hat{Z}_t^{(1)} = r_t + (1-d_t)\gamma Z_\theta^\pi(s_{t+1},a'_{t+1})$.
    Set the current weight $w = 1 - \lambda$.
    Combine the two targets, $(\hat{Z}_t^{(1)}, w)$ and $(\hat{Z}_t^{(\text{tot})}, w_{\text{tot}})$, and sort the combined target according to the positions of atoms.
    Build the CDF of the combined target by accumulating the weights at each atom.
    Project the combined target into a quantile distribution with $M'$ atoms, which is $\hat{Z}_t^{(\text{proj})}$, using the CDF (find the atom positions corresponding to each quantile).
    Update $\hat{Z}_{t-1}^{(\text{tot})} = r_{t-1} + (1-d_{t-1})\gamma\hat{Z}_t^{(\text{proj})}$ and $w_{\text{tot}} = \lambda\frac{\pi_\psi(a_t|s_t)}{\mu(a_t|s_t)}(1-d_{t-1})(1-\lambda+w_{\text{tot}})$.
**end for**
**Return** $\{\hat{Z}_t^{(\text{proj})}\}_{t=1}^{T}$.

---

### A.6 Quantitative Analysis on TD($\lambda$) Target Distribution

We experiment with a toy example to measure the bias and variance of the reward estimation according to $\lambda$. The toy example has two states, $s_1$ and $s_2$; the state distribution is defined as an uniform; the reward function is defined as $r(s_1) \sim \mathcal{N}(-0.005, 0.02)$ and $r(s_2) \sim \mathcal{N}(0.005, 0.03)$. We train parameterized reward distributions by minimizing the quantile regression loss with the TD($\lambda$) target distribution for $\lambda = 0, 0.5, 0.9$, and $1.0$. The experimental results are presented in the table below.

Table 1: Experimental results of the toy example.

|  | 5th iteration | 10th iteration | 15th iteration | 20th iteration | 25th iteration |
|---|---|---|---|---|---|
| $\lambda = 0.0$ | 4.813 (0.173) | 4.024 (0.253) | 3.498 (0.085) | 3.131 (0.103) | 2.835 (0.070) |
| $\lambda = 0.5$ | 4.621 (0.185) | 3.688 (0.273) | 2.925 (0.183) | 2.379 (0.134) | 2.057 (0.070) |
| $\lambda = 0.9$ | 4.141 (0.461) | 2.237 (0.402) | 1.389 (0.132) | 1.058 (0.031) | 0.923 (0.019) |
| $\lambda = 1.0$ | 2.886 (0.767) | 1.733 (0.365) | 1.509 (0.514) | 1.142 (0.325) | 1.109 (0.476) |

The values in the table are the mean and standard deviation of the past five values of the Wasserstein distance between the true reward return and the estimated distribution. Looking at the fifth iteration, it is clear that the larger the $\lambda$ value, the smaller the mean and the higher the standard deviation. At the 25th iteration, the run with $\lambda = 0.9$ has the lowest mean and standard deviation, indicating that training has converged. On the other hand, the run with $\lambda = 1.0$ has the biggest standard deviation, and the mean is greater than $\lambda = 0.9$, indicating that the significant variance hinders training. In conclusion, we measured bias and variance quantitatively through the toy example, and the results are well aligned with our claim that $\lambda$ can trade off bias and variance.

### A.7 Surrogate Functions

In this section, we introduce the surrogate functions for the objective and constraints. First, Kim and Oh [2022a] define a doubly discounted state distribution: $d_2^\pi(s) := (1 - \gamma^2) \sum_{t=0}^\infty \gamma^{2t} \Pr(s_t = s | \pi)$. Then, the surrogates for the objective and constraints are defined as follows [Kim and Oh, 2022a]:

$$J^{\mu,\pi}(\pi') := \underset{s_0 \sim \rho}{\mathbb{E}} [V^\pi(s_0)] + \frac{1}{1-\gamma} \left( \beta \underset{d^\pi}{\mathbb{E}} [H(\pi'(\cdot|s))] + \underset{d^{\mu,\pi'}}{\mathbb{E}} [Q_R^\pi(s,a)] \right),$$

$$J_{C_k}^{\mu,\pi}(\pi') := \underset{s_0 \sim \rho}{\mathbb{E}} [V_{C_k}^\pi(s_0)] + \frac{1}{1-\gamma} \underset{d^{\mu,\pi'}}{\mathbb{E}} [Q_{C_k}^\pi(s,a)],$$

$$J_{S_k}^{\mu,\pi}(\pi') := \underset{s_0 \sim \rho}{\mathbb{E}} [S_{C_k}^\pi(s_0)] + \frac{1}{1-\gamma^2} \underset{d_2^{\mu,\pi'}}{\mathbb{E}} [S_{C_k}^\pi(s,a)], \tag{37}$$

$$F_k^{\mu,\pi}(\pi'; \alpha) := J_{C_k}^{\mu,\pi}(\pi') + \frac{\phi(\Phi^{-1}(\alpha))}{\alpha} \sqrt{J_{S_k}^{\mu,\pi}(\pi') - (J_{C_k}^{\mu,\pi}(\pi'))^2},$$

where $\mu, \pi, \pi'$ are behavioral, current, and next policies, respectively. According to Theorem 1 in [Kim and Oh, 2022a], the constraint surrogates are bounded by $D_{\mathrm{KL}}(\pi, \pi')$. We also show that the surrogate of the objective is bounded by $D_{\mathrm{KL}}(\pi, \pi')$ in Appendix A.9. As a result, the gradients of the objective function and constraints become the same as the gradients of the surrogates, and the surrogates can substitute the objective and constraints within the trust region.

### A.8 Policy Update Rule

To solve the constrained optimization problem (5), we find a policy update direction by linearly approximating the objective and safety constraints and quadratically approximating the trust region constraint, as done by Achiam et al. [2017]. After finding the direction, we update the policy using a line search method. Given the current policy parameter $\psi_t \in \Psi$, the approximated problem can be expressed as follows:

$$x^* = \underset{x \in \Psi}{\arg\max} \, g^T x \quad \text{s.t.} \ \frac{1}{2} x^T H x \leq \epsilon, \ b_k^T x + c_k \leq 0 \ \forall k, \tag{38}$$

where $g = \nabla_\psi J^{\mu,\pi}(\pi_\psi)|_{\psi=\psi_t}$, $H = \nabla_\psi^2 D_{\mathrm{KL}}(\pi_{\psi_t} || \pi_\psi)|_{\psi=\psi_t}$, $b_k = \nabla_\psi F_k^{\mu,\pi}(\pi_\psi; \alpha)|_{\psi=\psi_t}$, and $c_k = F_k(\pi_\psi; \alpha) - d_k$. Since (38) is convex, we can use an existing convex optimization solver.

However, the search space, which is the policy parameter space $\Psi$, is excessively large, so we reduce the space by converting (38) to a dual problem as follows:

$$g(\lambda, \nu) = \min_x L(x, \lambda, \nu) = \min_x \{-g^T x + \nu(\frac{1}{2} x^T H x - \epsilon) + \lambda^T (Bx + c)\}$$

$$= \frac{-1}{2\nu}\left(\underbrace{g^T H^{-1} g}_{=:q} - 2\underbrace{g^T H^{-1} B^T}_{=:r^T}\lambda + \lambda^T \underbrace{BH^{-1}B^T}_{=:S}\lambda\right) + \lambda^T c - \nu\epsilon \qquad (39)$$

$$= \frac{-1}{2\nu}(q - 2r^T\lambda + \lambda^T S\lambda) + \lambda^T c - \nu\epsilon,$$

where $B = (b_1, .., b_K)$, $c = (c_1, ..., c_K)^T$, and $\lambda \in \mathbb{R}^K \geq 0$ and $\nu \in \mathbb{R} \geq 0$ are Lagrange multipliers. Then, the optimal $\lambda$ and $\nu$ can be obtained by a convex optimization solver. After obtaining the optimal values, $(\lambda^*, \nu^*) = \operatorname{argmax}_{(\lambda,\nu)} g(\lambda, \nu)$, the policy update direction $x^*$ are calculated by $\frac{1}{\nu^*} H^{-1}(g - B^T\lambda^*)$. Then, the policy is updated by $\psi_{t+1} = \psi_t + \beta x^*$, where $\beta$ is a step size, which can be found through a backtracking method (please refer to Section 6.3.2 of Dennis and Schnabel [1996]).

Before using the above policy update rule, we should note that the existing trust-region method with the risk-averse constraint [Kim and Oh, 2022a] and the equations (1, 37, 5) are slightly different. There are two differences: 1) the objective is augmented with an entropy bonus, and 2) the surrogates are expressed with Q-functions instead of value functions. To use the entropy-regularized objective in the trust-region method, it is required to show that the objective is bounded by the KL divergence. We present the existence of bound in Appendix A.9. Next, there is no problem using the Q-functions because it is mathematically equivalent between the original surrogates [Kim and Oh, 2022a] and the new ones expressed with Q-functions (37). However, we experimentally show that using the Q-functions in off-policy settings has advantages in Appendix A.10.

## A.9  Bound of Entropy-Augmented Objective

In this section, we show that the entropy-regularzied objective in (1) has a bound expressed by the KL divergence. Before showing the boundness, we present a new function and a lemma. A value difference function is defined as follows:

$$\delta^{\pi'}(s) := \mathbb{E}\left[R(s, a, s') + \gamma V^\pi(s') - V^\pi(s) \mid a \sim \pi'(\cdot|s), s' \sim P(\cdot|s, a)\right] = \mathbb{E}_{a \sim \pi'}\left[A^\pi(s, a)\right],$$

where $A^\pi(s, a) := Q^\pi(s, a) - V^\pi(s, a)$.

**Lemma A.7.** *The maximum of $|\delta^{\pi'}(s) - \delta^\pi(s)|$ is equal or less than $\epsilon_R\sqrt{2D_{\mathrm{KL}}^{\max}(\pi||\pi')}$, where $\epsilon_R = \max_{s,a}|A^\pi(s, a)|$.*

*Proof.* The value difference can be expressed in a vector form,

$$\delta^{\pi'}(s) - \delta^\pi(s) = \sum_a (\pi'(a|s) - \pi(a|s))A^\pi(s, a) = \langle \pi'(\cdot|s) - \pi(\cdot|s), A^\pi(s, \cdot)\rangle.$$

Using Hölder's inequality, the following inequality holds:

$$|\delta^{\pi'}(s) - \delta^\pi(s)| \leq ||\pi'(\cdot|s) - \pi(\cdot|s)||_1 \cdot ||A^\pi(s, \cdot)||_\infty$$
$$= 2D_{\mathrm{TV}}(\pi'(\cdot|s)||\pi(\cdot|s))\max_a A^\pi(s, a).$$
$$\Rightarrow ||\delta^{\pi'} - \delta^\pi||_\infty = \max_s|\delta^{\pi'}(s) - \delta^\pi(s)| \leq 2\epsilon_R\max_s D_{\mathrm{TV}}(\pi(\cdot|s)||\pi'(\cdot|s)).$$

Using Pinsker's inequality, $||\delta^{\pi'} - \delta^\pi||_\infty \leq \epsilon_R\sqrt{2D_{\mathrm{KL}}^{\max}(\pi||\pi')}$. $\qquad\square$

**Theorem A.8.** *Let us assume that $\max_s H(\pi(\cdot|s)) < \infty$ for $\forall\pi \in \Pi$. The difference between the objective and surrogate functions is bounded by a term consisting of KL divergence as:*

$$\left|J(\pi') - J^{\mu,\pi}(\pi')\right| \leq \frac{\sqrt{2}\gamma}{(1-\gamma)^2}\sqrt{D_{\mathrm{KL}}^{\max}(\pi||\pi')}\left(\beta\epsilon_H + \epsilon_R\sqrt{2D_{\mathrm{KL}}^{\max}(\mu||\pi')}\right), \qquad (40)$$

*where $\epsilon_H = \max_s |H(\pi'(\cdot|s))|$, $D_{\mathrm{KL}}^{\max}(\pi||\pi') = \max_s D_{\mathrm{KL}}(\pi(\cdot|s)||\pi'(\cdot|s))$, and the equality holds when $\pi' = \pi$.*

*Proof.* The surrogate function can be expressed in vector form as follows:

$$J^{\mu,\pi}(\pi') = \langle \rho, V^\pi \rangle + \frac{1}{1-\gamma}\left(\langle d^\mu, \delta^{\pi'} \rangle + \beta\langle d^\pi, H^{\pi'} \rangle\right),$$

where $H^{\pi'}(s) = H(\pi'(\cdot|s))$. The objective function of $\pi'$ can also be expressed in a vector form using Lemma 1 from Achiam et al. [2017],

$$\begin{aligned}
J(\pi') &= \frac{1}{1-\gamma}\mathbb{E}\left[R(s,a,s') + \beta H^{\pi'}(s) \mid s \sim d^{\pi'}, a \sim \pi'(\cdot|s), s' \sim P(\cdot|s,a)\right] \\
&= \frac{1}{1-\gamma}\mathop{\mathbb{E}}_{s \sim d^{\pi'}}\left[\delta^{\pi'}(s) + \beta H^{\pi'}(s)\right] + \mathop{\mathbb{E}}_{s \sim \rho}\left[V^\pi(s)\right] \\
&= \langle \rho, V^\pi \rangle + \frac{1}{1-\gamma}\langle d^{\pi'}, \delta^{\pi'} + \beta H^{\pi'} \rangle.
\end{aligned}$$

By Lemma 3 from Achiam et al. [2017], $||d^\pi - d^{\pi'}||_1 \leq \frac{\gamma}{1-\gamma}\sqrt{2D_{\mathrm{KL}}^{\max}(\pi||\pi')}$. Then, the following inequality is satisfied:

$$\begin{aligned}
|(1-\gamma)&(J^{\mu,\pi}(\pi') - J(\pi'))| \\
&= |\langle d^{\pi'} - d^\mu, \delta^{\pi'} \rangle + \beta\langle d^\pi - d^{\pi'}, H^{\pi'} \rangle| \\
&\leq |\langle d^{\pi'} - d^\mu, \delta^{\pi'} \rangle| + \beta|\langle d^\pi - d^{\pi'}, H^{\pi'} \rangle| \\
&= |\langle d^{\pi'} - d^\mu, \delta^{\pi'} - \delta^\pi \rangle| + \beta|\langle d^\pi - d^{\pi'}, H^{\pi'} \rangle| && (\because \delta^\pi = 0) \\
&\leq ||d^{\pi'} - d^\mu||_1 ||\delta^{\pi'} - \delta^\pi||_\infty + \beta||d^\pi - d^{\pi'}||_1 ||H^{\pi'}||_\infty && (\because \text{Hölder's inequality}) \\
&\leq \frac{2\epsilon_R\gamma}{1-\gamma}\sqrt{D_{\mathrm{KL}}^{\max}(\mu||\pi')D_{\mathrm{KL}}^{\max}(\pi||\pi')} + \frac{\beta\gamma\epsilon_H}{1-\gamma}\sqrt{2D_{\mathrm{KL}}^{\max}(\pi||\pi')} && (\because \text{Lemma A.7}) \\
&= \frac{\gamma}{1-\gamma}\sqrt{D_{\mathrm{KL}}^{\max}(\pi||\pi')}\left(\sqrt{2}\beta\epsilon_H + 2\epsilon_R\sqrt{D_{\mathrm{KL}}^{\max}(\mu||\pi')}\right).
\end{aligned}$$

If $\pi' = \pi$, the KL divergence term becomes zero, so equality holds. $\qquad\square$

### A.10  Comparison of Q-Function and Value Function-Based Surrogates

The original surrogate is defined as follows:

$$J^{\mu,\pi}(\pi') := J(\pi) + \frac{1}{1-\gamma}\mathop{\mathbb{E}}_{d^\mu,\mu}\left[\frac{\pi'(a|s)}{\mu(a|s)}A^\pi(s,a)\right], \tag{41}$$

where $A^\pi(s,a) := Q^\pi(s,a) - V^\pi(s)$, and the surrogate is the same as that of OffTRPO [Meng et al., 2022] and OffTRC [Kim and Oh, 2022a]. An entropy-regularized version can be derived as:

$$J^{\mu,\pi}(\pi') = J(\pi) + \frac{1}{1-\gamma}\left(\beta\mathop{\mathbb{E}}_{d^\pi}\left[H(\pi'(\cdot|s))\right] + \mathop{\mathbb{E}}_{d^\mu,\mu}\left[\frac{\pi'(a|s)}{\mu(a|s)}A^\pi(s,a)\right]\right). \tag{42}$$

Then, the surrogate expressed by Q-functions in (37), called SAC-style version, can be rewritten as:

$$J^{\mu,\pi}(\pi') = J(\pi) + \frac{1}{1-\gamma}\left(\beta\mathop{\mathbb{E}}_{d^\pi}\left[H(\pi'(\cdot|s))\right] + \mathop{\mathbb{E}}_{d^\mu,\pi'}\left[Q^\pi(s,a)\right]\right). \tag{43}$$

In this section, we evaluate the original, entropy-regularized, and SAC-style versions in the continuous control tasks of the MuJoCo simulators [Todorov et al., 2012]. We use neural networks with two hidden layers with (512, 512) nodes and ReLU for the activation function. The output of a value network is linear, but the input is different; the original and entropy-regularized versions use states, and the SAC-style version uses state-action pairs. The input of a policy network is the state, the output is mean $\mu$ and std $\sigma$, and actions are squashed into $\tanh(\mu + \epsilon\sigma)$, $\epsilon \sim \mathcal{N}(0,1)$ as in SAC [Haarnoja et al., 2018]. The entropy coefficient $\beta$ in the entropy-regularized and SAC-style versions are adaptively adjusted to keep the entropy above a threshold (set as $-d$ given $A \subseteq \mathbb{R}^d$). The hyperparameters for all versions are summarized in Table 2.

The training curves are presented in Figure 5. All methods are trained with five different random seeds. Although the entropy-regularized version (42) and SAC-style version (43) are mathematically

Table 2: Hyperparameters for all versions.

| Parameter | Value |
|---|---|
| Discount factor $\gamma$ | 0.99 |
| Trust region size $\epsilon$ | 0.001 |
| Length of replay buffer | $10^5$ |
| Critic learning rate | 0.0003 |
| Trace-decay $\lambda$ | 0.97 |
| Initial entropy coefficient $\beta$ | 1.0 |
| $\beta$ learning rate | 0.01 |

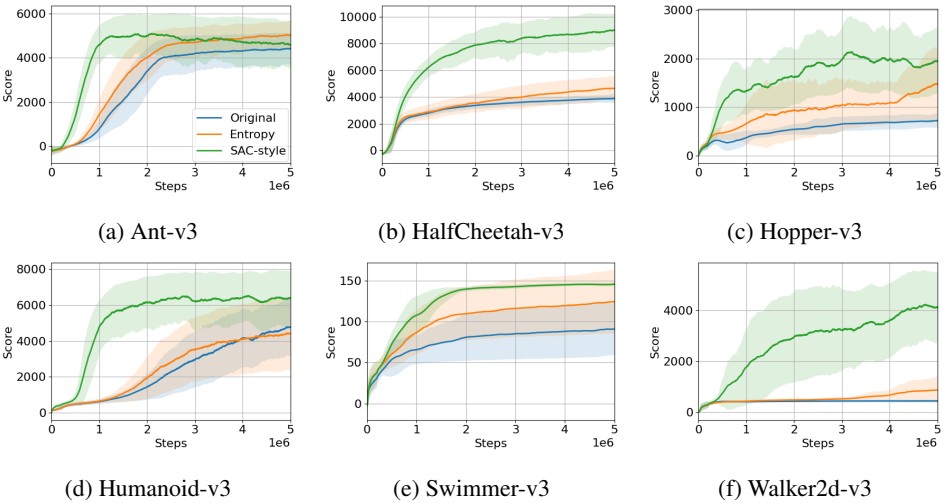

(a) Ant-v3      (b) HalfCheetah-v3      (c) Hopper-v3

(d) Humanoid-v3      (e) Swimmer-v3      (f) Walker2d-v3

Figure 5: MuJoCo training curves.

equivalent, it can be observed that the performance of the SAC-style version is superior to the regularized version. It can be inferred that this is due to the variance of importance sampling. In the off-policy setting, the sampling probabilities of the behavioral and current policies can be significantly different, so the variance of the importance ratio is huge. The increased variance prevents estimating the objective accurately, so significant performance degradation can happen. As a result, using the Q-function-based surrogates has an advantage for efficient learning.

# B Experimental Settings

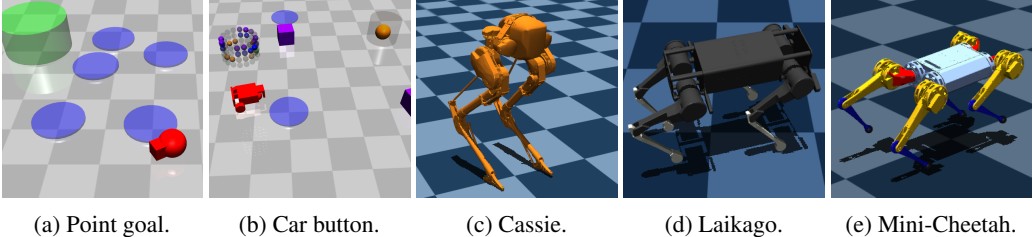

(a) Point goal.      (b) Car button.      (c) Cassie.      (d) Laikago.      (e) Mini-Cheetah.

Figure 6: (a) and (b) are Safety Gym tasks. (c), (d), and (e) are locomotion tasks.

**Safety Gym.** We use the goal and button tasks with the point and car robots in the Safety Gym environment [Ray et al., 2019], as shown in Figure 6a and 6b. The environmental setting for the goal task is the same as in Kim and Oh [2022b]. Eight hazard regions and one goal are randomly spawned at the beginning of each episode, and a robot gets a reward and cost as follows:

$$R(s, a, s') = -\Delta d_{\text{goal}} + \mathbf{1}_{d_{\text{goal}} \leq 0.3},$$
$$C(s, a, s') = \text{Sigmoid}(10 \cdot (0.2 - d_{\text{hazard}})), \tag{44}$$

where $d_{\text{goal}}$ is the distance to the goal, and $d_{\text{hazard}}$ is the minimum distance to hazard regions. If $d_{\text{goal}}$ is less than or equal to $0.3$, a goal is respawned. The state consists of relative goal position, goal distance, linear and angular velocities, acceleration, and LiDAR values. The action space is two-dimensional, which consists of $xy$-directional forces for the point and wheel velocities for the car robot.

The environmental settings for the button task are the same as in Liu et al. [2022]. There are five hazard regions, four dynamic obstacles, and four buttons, and all components are fixed throughout the training. The initial position of a robot and an activated button are randomly placed at the beginning of each episode. The reward function is the same as in (44), but the cost is different since there is no dense signal for contacts. We define the cost function for the button task as an indicator function that outputs one if the robot makes contact with an obstacle or an inactive button or enters a hazardous region. We add LiDAR values of buttons and obstacles to the state of the goal task, and actions are the same as the goal task. The length of the episode is 1000 steps without early termination.

**Locomotion Tasks.** We use three different legged robots, Mini-Cheetah, Laikago, and Cassie, for the locomotion tasks, as shown in Figure 6e, 6d, and 6c. The tasks aim to control robots to follow a velocity command on flat terrain. A velocity command is given by $(v_x^{\text{cmd}}, v_y^{\text{cmd}}, \omega_z^{\text{cmd}})$, where $v_x^{\text{cmd}} \sim \mathcal{U}(-1.0, 1.0)$ for Cassie and $\mathcal{U}(-1.0, 2.0)$ otherwise, $v_y^{\text{cmd}} = 0$, and $\omega_z^{\text{cmd}} \sim \mathcal{U}(-0.5, 0.5)$. To lower the task complexity, we set the $y$-directional linear velocity to zero but can scale to any non-zero value. As in other locomotion studies [Lee et al., 2020, Miki et al., 2022], *central phases* are introduced to produce periodic motion, which are defined as $\phi_i(t) = \phi_{i,0} + f \cdot t$ for $\forall i \in \{1, ..., n_{\text{legs}}\}$, where $f$ is a frequency coefficient and is set to 10, and $\phi_{i,0}$ is an initial phase. Actuators of robots are controlled by PD control towards target positions given by actions. The state consists of velocity command, orientation of the robot frame, linear and angular velocities of the robot, positions and speeds of the actuators, central phases, history of positions and speeds of the actuators (past two steps), and history of actions (past two steps). A foot contact timing $\xi$ can be defined as follows:

$$\xi_i(s) = -1 + 2 \cdot \mathbf{1}_{\sin(\phi_i) \leq 0} \quad \forall i \in \{1, ..., n_{\text{legs}}\}, \tag{45}$$

where a value of -1 means that the $i$th foot is on the ground; otherwise, the foot is in the air. For the quadrupedal robots, Mini-Cheetah and Laikago, we use the initial phases as $\phi_0 = \{0, \pi, \pi, 0\}$, which generates trot gaits. For the bipedal robot, Cassie, the initial phases are defined as $\phi_0 = \{0, \pi\}$, which generates walk gaits. Then, the reward and cost functions are defined as follows:

$$R(s, a, s') = -0.1 \cdot (||v_{x,y}^{\text{base}} - v_{x,y}^{\text{cmd}}||_2^2 + ||\omega_z^{\text{base}} - \omega_z^{\text{cmd}}||_2^2 + 10^{-3} \cdot R_{\text{power}}),$$
$$C_1(s, a, s') = \mathbf{1}_{\text{angle} \geq a}, \ C_2(s, a, s') = \mathbf{1}_{\text{height} \leq b}, \ C_3(s, a, s') = \sum_{i=1}^{n_{\text{legs}}} (1 - \xi_i \cdot \hat{\xi}_i)/(2 \cdot n_{\text{legs}}), \tag{46}$$

where the power consumption $R_{\text{power}} = \sum_i |\tau_i v_i|$, the sum of the torque times the actuator speed, is added to the reward as a regularization term, $v_{x,y}^{\text{base}}$ is the $xy$-directional linear velocity of the base

frame of robots, $\omega_z^{\text{base}}$ is the $z$-directional angular velocity of the base frame, and $\hat{\xi} \in \{-1, 1\}^{n_{\text{legs}}}$ is the current feet contact vector. For balancing, the first cost indicates whether the angle between the $z$-axis vector of the robot base and the world is greater than a threshold ($a = 15°$ for all robots). For standing, the second cost indicates the height of CoM is less than a threshold ($b = 0.3, 0.35, 0.7$ for Mini-Cheetah, Laikago, and Cassie, respectively), and the last cost is to check that the current feet contact vector $\hat{\xi}$ matches the pre-defined timing $\xi$. The length of the episode is 500 steps. There is no early termination, but if a robot falls to the ground, the state is frozen until the end of the episode.

**Hyperparameter Settings.** The structure of neural networks consists of two hidden layers with $(512, 512)$ nodes and ReLU activation for all baselines and the proposed method. The input of value networks is state-action pairs, and the output is the positions of atoms. The input of policy networks is the state, the output is mean $\mu$ and std $\sigma$, and actions are squashed into $\tanh(\mu + \epsilon\sigma)$, $\epsilon \sim \mathcal{N}(0, 1)$. We use a fixed entropy coefficient $\beta$. The trust region size $\epsilon$ is set to $0.001$ for all trust region-based methods. The overall hyperparameters for the proposed method can be summarized in Table 3.

Table 3: Hyperparameter settings for the Safety Gym and locomotion tasks.

| Parameter | Safety Gym | Locomotion |
|---|---|---|
| Discount factor $\gamma$ | 0.99 | 0.99 |
| Trust region size $\epsilon$ | 0.001 | 0.001 |
| Length of replay buffer | $10^5$ | $10^5$ |
| Critic learning rate | 0.0003 | 0.0003 |
| Trace-decay $\lambda$ | 0.97 | 0.97 |
| Entropy coefficient $\beta$ | 0.0 | 0.001 |
| The number of critic atoms $M$ | 25 | 25 |
| The number of target atoms $M'$ | 50 | 50 |
| Constraint risk level $\alpha$ | 0.25, 0.5, and 1.0 | 1.0 |
| threshold $d_k$ | $0.025/(1-\gamma)$ | $[0.025, 0.025, 0.4]/(1-\gamma)$ |
| Slack coefficient $\zeta$ | - | $\min_k d_k = 0.025/(1-\gamma)$ |

Since the range of the cost is $[0, 1]$, the maximum discounted cost sum is $1/(1-\gamma)$. Thus, the threshold is set by target cost rate times $1/(1-\gamma)$. For the locomotion tasks, the third cost in (46) is designed for foot stamping, which is not essential to safety. Hence, we set the threshold to near the maximum (if a robot does not stamp, the cost rate becomes 0.5). In addition, baseline safe RL methods use multiple critic networks for the cost function, such as target [Yang et al., 2021] or square value networks [Kim and Oh, 2022a]. To match the number of network parameters, we use two critics as an ensemble, as in Kuznetsov et al. [2020].

**Tips for Hyperparameter Tuning.**

- Discount factor $\gamma$, Critic learning rate: Since these are commonly used hyperparameters, we do not discuss these.

- Trace-decay $\lambda$, Trust region size $\epsilon$: The ablation studies on these hyperparameters are presented in Appendix C.3. From the results, we recommend setting the trace-decay to $0.95 \sim 0.99$ as in other TD($\lambda$)-based methods [Precup et al., 2000]. Also, the results show that the performance is not sensitive to the trust region size. However, if the trust region size is too large, the approximation error increases, so it is better to set it below $0.003$.

- Entropy coefficient $\beta$: This value is fixed in our experiments, but it can be adjusted automatically as done in SAC [Haarnoja et al., 2018].

- The number of atoms $M, M'$: Although experiments on the number of atoms did not performed, performance is expected to increase as the number of atoms increases, as in other distributional RL methods [Dabney et al., 2018a].

- Length of replay buffer: The effect of the length of the replay buffer can be confirmed through the experimental results from an off policy-based safe RL method [Kim and Oh, 2022a]. According to that, the length does not impact performance unless it is too short. We recommend setting it to 10 to 100 times the collected trajectory length.

- Constraint risk level $\alpha$, threshold $d_k$: If the cost sum follows a Gaussian distribution, the mean-std constraint is identical to the CVaR constraint. Then, the probability of the

worst case can be controlled by adjusting $\alpha$. For example, if we set $\alpha = 0.125$ and $d = 0.03/(1 - \gamma)$, the mean-std constraint enforces the probability that the average cost is less than 0.03 during an episode greater than $95\% = \Phi(\phi(\Phi^{-1}(\alpha))/\alpha)$. Through this meaning, proper $\alpha$ and $d_k$ can be found.

- Slack coefficient $\zeta$: As mentioned at the end of Section 3.1, it is recommended to set this coefficient as large as possible. Since $d_k - \zeta$ should be positive, we recommend setting $\zeta$ to $\min_k d_k$.

In conclusion, most hyperparameters are not sensitive, so few need to be optimized. It seems that $\alpha$ and $d_k$ need to be set based on the meaning described above. Additionally, if the approximation error of critics is significant, the trust region size should be set smaller.

# C  Experimental Results

## C.1  Safety Gym

In this section, we present the training curves of the Safety Gym tasks separately according to the risk level of constraints for better readability. Figure 7 shows The training results of the risk-neutral constrained algorithms and risk-averse constrained algorithms with $\alpha = 1.0$. Figures 8 and 9 show the training results of the risk-averse constrained algorithms with $\alpha = 0.25$ and $0.5$, respectively.

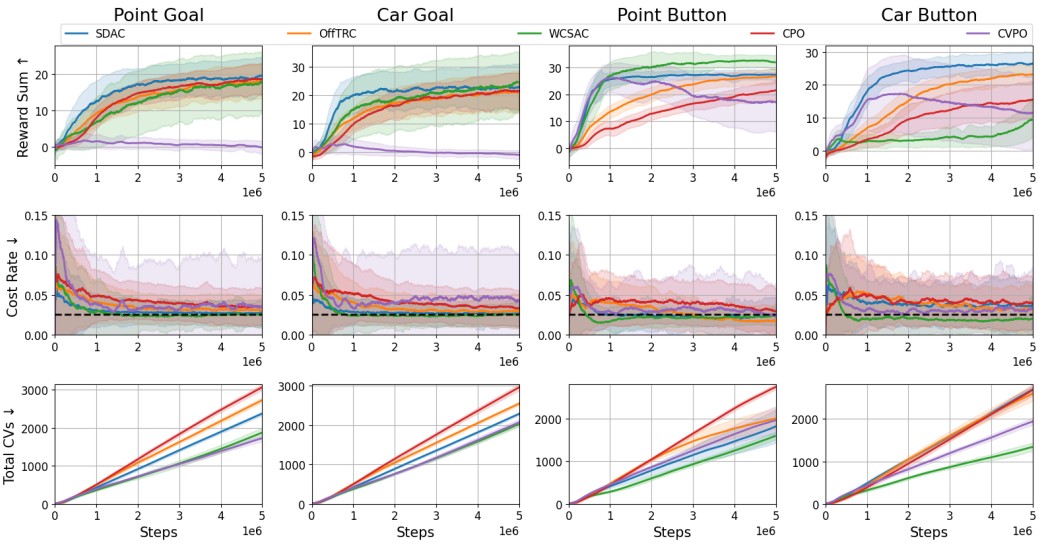

Figure 7: Training curves of risk-neutral constrained algorithms for the Safety Gym tasks. The solid line and shaded area represent the average and std values, respectively. The black dashed lines in the second row indicate thresholds. All methods are trained with five random seeds.

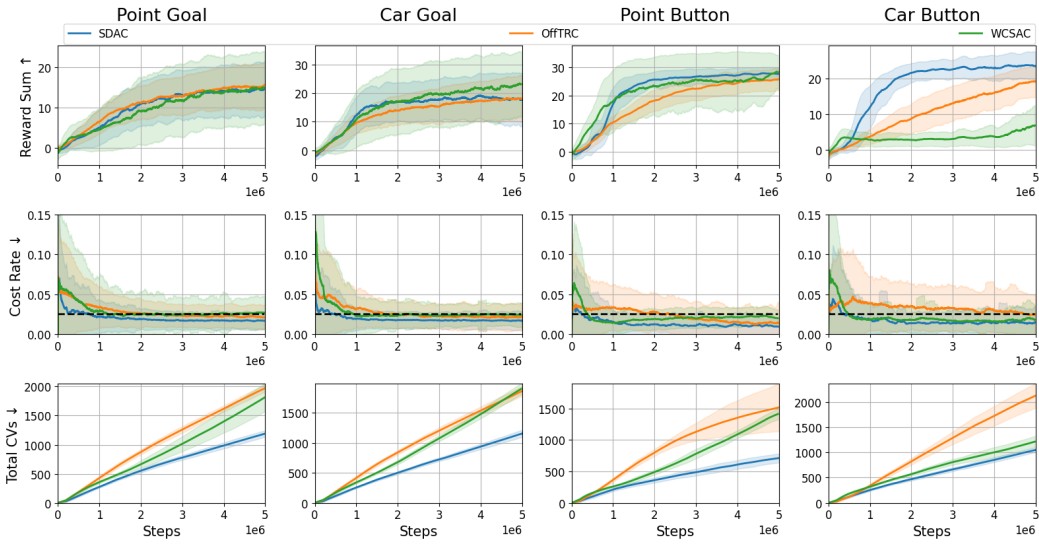

Figure 8: Training curves of risk-averse constrained algorithms with $\alpha = 0.5$ for the Safety Gym.

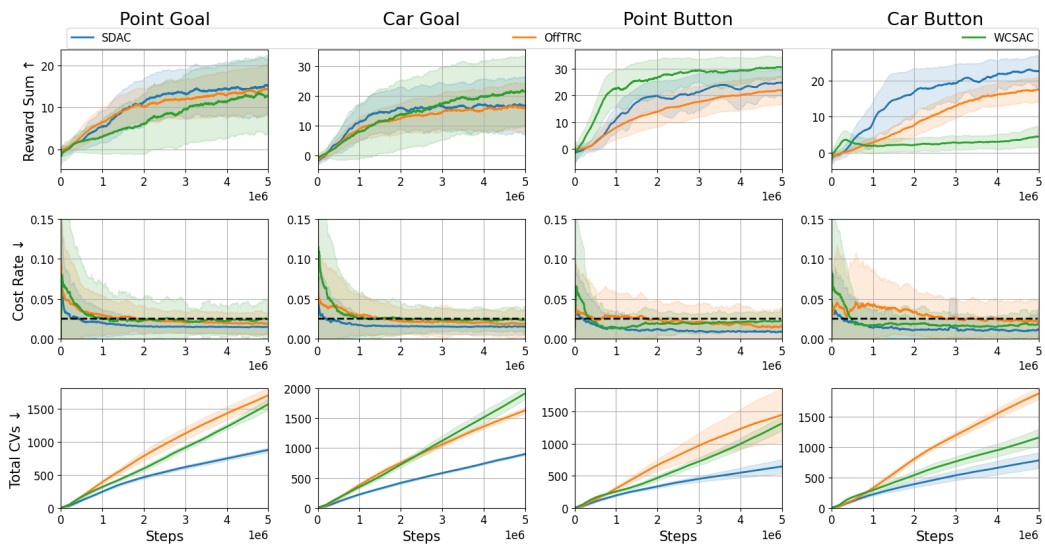

Figure 9: Training curves of risk-averse constrained algorithms with $\alpha = 0.25$ for the Safety Gym.

## C.2 Ablation Study on Components of SDAC

There are three main differences between SDAC and the existing trust region-based safe RL algorithm for mean-std constraints [Kim and Oh, 2022a], called OffTRC: 1) feasibility handling methods in multi-constraint settings, 2) the use of distributional critics, and 3) the use of Q-functions instead of advantage functions, as explained in Appendix A.8 and A.10. Since the ablation study for feasibility handling is conducted in Section 5.3, we perform ablation studies for the distributional critic and Q-function in this section. We call SDAC with only distributional critics as *SDAC-Dist* and SDAC with only Q-functions as *SDAC-Q*. If all components are absent, SDAC is identical to OffTRC [Kim and Oh, 2022a]. The variants are trained with the point goal task of the Safety Gym, and the training results are shown in Figure 10. SDAC-Q lowers the cost rate quickly but shows the lowest score. SDAC-Dist shows scores similar to SDAC, but the cost rate converges above the threshold $0.025$. In conclusion, SDAC can efficiently satisfy the safety constraints through the use of Q-functions and improve score performance through the distributional critics.

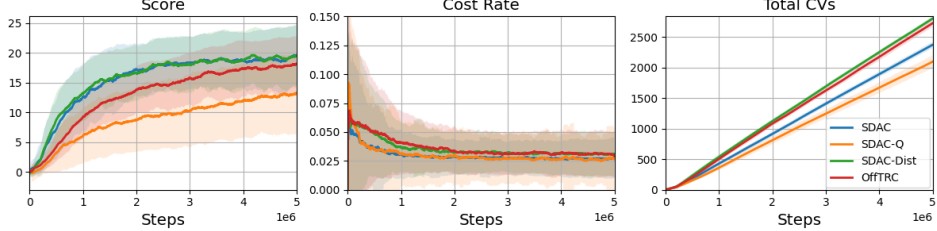

Figure 10: Training curves of variants of SDAC for the point goal task.

## C.3   Ablation Study on Hyperparameters

To check the effects of the hyperparameters, we conduct ablation studies on the trust region size $\epsilon$ and entropy coefficient $\beta$. The results on the entropy coefficient are presented in Figure 11a, showing that the score significantly decreases when $\beta$ is $0.01$. This indicates that policies with high entropy fail to improve score performance since they focus on satisfying the constraints. Thus, the entropy coefficient should be adjusted cautiously, or it can be better to set the coefficient to zero. The results on the trust region size are shown in Figure 11b, which shows that the results do not change significantly regardless of the trust region size. However, the score convergence rate for $\epsilon = 0.01$ is the slowest because the estimation error of the surrogate increases as the trust region size increases according to Theorem A.8.

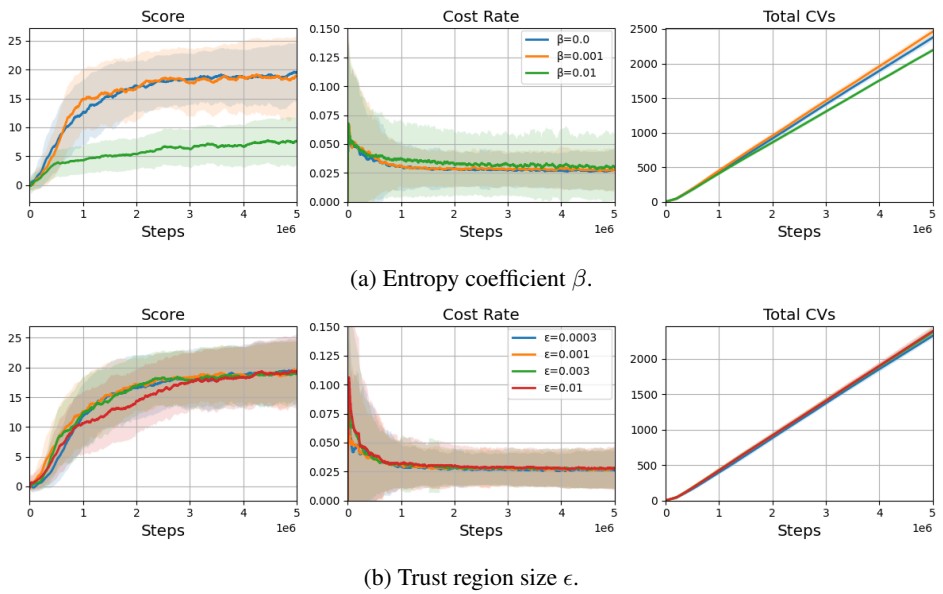

(a) Entropy coefficient $\beta$.

(b) Trust region size $\epsilon$.

Figure 11: Training curves of SDAC with different hyperparameters for the point goal task.

# D Comparison with RL Algorithms

In this section, we compare the proposed safe RL algorithm with traditional RL algorithms in the locomotion tasks and show that safe RL has the advantage of not requiring reward tuning. We use the truncated quantile critic (TQC) [Kuznetsov et al., 2020], a state-of-the-art algorithm in existing RL benchmarks [Todorov et al., 2012], as traditional RL baselines. To apply the same experiment to traditional RL, it is necessary to design a reward reflecting safety. We construct the reward through a weighted sum as $\bar{R} = (R - \sum_{i=1}^{3} w_i C_i)/(1 + \sum_{i=1}^{3} w_i)$, where $R$ and $C_{\{1,2,3\}}$ are used to train safe RL methods and are defined in Appendix B, and $R$ is called the *true reward*. The weights of the reward function $w_{\{1,2,3\}}$ are searched by a Bayesian optimization tool[1] to maximize the true reward of TQC in the Mini-Cheetah task. Among the 63 weights searched through Bayesian optimization, the top five weights are listed in Table 4.

Table 4: Weights of the reward function for the Mini-Cheetah task.

| Reward weights | $w_1$ | $w_2$ | $w_3$ |
|---|---|---|---|
| #1 | 1.588 | 0.299 | 0.174 |
| #2 | 1.340 | 0.284 | 0.148 |
| #3 | 1.841 | 0.545 | 0.951 |
| #4 | 6.560 | 0.187 | 4.920 |
| #5 | 1.603 | 0.448 | 0.564 |

Figure 12 shows the training curves of the Mini-Cheetah task experiments where TQC is trained using the weight pairs listed in Table 4. The graph shows that it is difficult for TQC to lower the second cost below the threshold while all costs of SDAC are below the threshold. In particular, TQC with the fifth weight pairs shows the lowest second cost rate, but the true reward sum is the lowest. This shows that it is challenging to obtain good task performance while satisfying the constraints through reward tuning.

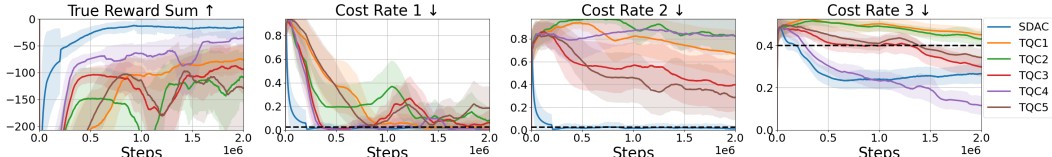

Figure 12: Training curves of the Mini-Cheetah task. The black dashed lines show the thresholds used for the safe RL method. The solid line represents the average value, and the shaded area shows one-fifth of the std value. The number after TQC in the legend indicates which of the reward weights in Table 4 is used. All methods are trained with five different random seeds.

---

[1]We use Sweeps from Weights & Biases Biewald [2020].

# E Computational Cost Analysis

## E.1 Complexity of Gradient Integration Method

In this section, we analyze the computational cost of the gradient integration method. The proposed gradient integration method has three subparts. First, it is required to calculate policy gradients of each cost surrogate, $g_k$, and $H^{-1}g_k$ for $\forall k \in \{1, 2, ..., K\}$, where $H$ is the Hessian matrix of the KL divergence. $H^{-1}g_k$ can be computed using the conjugate gradient method, which requires only a constant number of back-propagation on the cost surrogate, so the computational cost can be expressed as $K \cdot O(\text{BackProp})$.

Second, the quadratic problem in Section 3.1 is transformed to a dual problem, where the transformation process requires inner products between $g_k$ and $H^{-1}g_m$ for $\forall k, m \in \{1, 2, ..., K\}$. The computational cost can be expressed as $K^2 \cdot O(\text{InnerProd})$.

Finally, the transformed quadratic problem is solved in the dual space $\in \mathbb{R}^K$ using a quadratic programming solver. Since $K$ is usually much smaller than the number of policy parameters, the computational cost almost negligible compared to the others. Then, the cost of the gradient integration is $K \cdot O(\text{BackProp}) + K^2 \cdot O(\text{InnerProd}) + C$. Since the back-propagation and the inner products is proportional to the number of policy parameters $|\psi|$, the computational cost can be simplified as $O(K^2 \cdot |\psi|)$.

## E.2 Quantitative Analysis

Table 5: Training time of Safe RL algorithms (in hours). The training time of each algorithm is measured as the average time required for training with five random seeds. The total training steps are $5 \cdot 10^6$ and $3 \cdot 10^6$ for the point goal task and the Mini-Cheetah task, respectively.

| Task | SDAC (proposed) | OffTRC | WCSAC | CPO | CVPO |
|---|---|---|---|---|---|
| Point goal (Safety Gym) | 7.96 | 4.86 | 19.07 | 2.61 | 47.43 |
| Mini-Cheetah (Locomotion) | 8.36 | 6.54 | 16.41 | 1.99 | - |

We analyze the computational cost of the proposed method quantitatively. To do this, we measure the training time of the proposed method, SDAC, and the safe RL baselines. We use a workstation whose CPU is the Intel Xeon e5-2650 v3, and GPU is the NVIDIA GeForce GTX TITAN X. The results are presented in Table 5. While CPO is the fastest algorithm, its performance, such as the sum of rewards, is relatively poor compared to other algorithms. The main reason why CPO shows the fastest computation time is that CPO is an on-policy algorithm, hence, it does not require an insertion to (and deletion from) a replay memory, and batch sampling. SDAC shows the third fastest computation time in all algorithms and the second best one among off-policy algorithms. Especially, SDAC is slightly slower than OffTRC, which is the fastest one among off-policy algorithms. This result shows the benefit of SDAC since SDAC outperforms OffTRC in terms of the returns and CV, but the training time is not significantly increased over OffTRC. WCSAC, which is based on SAC, has a slower training time because it updates networks more frequently than other algorithms. CVPO, an EM-based safe RL algorithm, has the slowest training time. In the E-step of CVPO, a non-parametric policy is optimized to solve a local subproblem, and the optimization process requires discretizing the action space and solving a non-linear convex optimization for all batch states. Because of this, CVPO takes the longest to train an RL agent.

