# OpenReview forum: "Trust Region-Based Safe Distributional Reinforcement Learning for Multiple Constraints"
_NeurIPS.cc/2023/Conference — NeurIPS 2023 poster_

### Official Review · Reviewer_Cyuw · 2023-07-04

**Soundness:** 3 good
**Presentation:** 2 fair
**Contribution:** 2 fair
**Rating:** 6
**Confidence:** 3

**Summary:**

The paper proposed a trust-region method for handling multiple constraints in safe RL for CMDPs. If the TRPO-like gradient calculation with multiple constraints is not feasible, the authors proposed a gradient integration method to calculate feasible gradients. The authors also proposed a TD-$\lambda$ method to calculate the target distribution of distributional critic. Experimental results have shown that the proposed methods seem to outperform baselines.

**Strengths:**

## Originality
The gradient integration is novel and looks useful for the infeasible initial stage when solving CMDP. Computing the TD $\lambda$ for the distributional critic is also novel to me.

## Quality
The paper is well-written and includes many technical details, which is good for understanding and reproducing the work.

## Clarity
The paper is very clear. Especially, I like the two figures which explain the ideas very well and I get the idea immediately. The notation is a little bit messy but clear enough to read.

## Significance
The TD-$\lambda$ method for distributional critic might have a broad effect on other distributional RL algorithms, but I am not sure if similar approaches have been proposed to solve the problem.

**Weaknesses:**

1. The paper has novelty but the contributions are not clearly demonstrated. From the results in Figures 3 and 4, I did not see too much performance improvement, solid constraint satisfaction, or faster convergence to feasible policies compared to baselines. Given this paper focuses on practical contributions, the performance results are not convincing.

2. Two contributions of this paper are not related. Gradient integration and TD-$\lambda$ seem to stand alone on their own. The authors should explain why you put these into one paper and how they contribute together to improve the safe RL.

3. The gradient integration problem is fragile and might have some problems in the finite-convergence theorem. I will put more discussion in the questions section.

**Questions:**

1. I got confused about the figures of the first column in Figure 4. The curves and shades are not continuous. Please explain why.

*Updates after rebuttal: Clarified in the reply. They are random samples.*

2. For Figure 1, what if the $g_1$ and $g_2$ are parallel to each other (no feasible region), or have an arbitrarily small relative angle? Then you integrate them to near zero and get stuck. Then I doubt the conclusion about finite-time convergence to feasible region. You might need more assumptions like the feasible region does exist. Do you want to comment on these situations?

3. Following problems in 2, it seems like no guarantees like the proposed gradient integration could be faster than naive methods could be made. Then how exactly can safe RL methods benefit from the gradient integration?

3. Practically, Solving TRPO is already computationally heavy. Locally linearization of multiple constraints and objectives will cause more approximation and further harm the performance. If you only would like to handle multiple constraints, many straightforward approaches could be used like first optimizing a weighted sum of multiple constraint critics. How does the proposed algorithm outperform these intuitions then?

Please carefully explain these questions. I will consider increasing the score if the authors give me a reasonable response.

**Limitations:**

The authors addressed the limitations well.

---

> ### Author Rebuttal · Authors · 2023-08-08
>
> We thank reviewer Cyuw for the feedback and thorough review on our work.
> We respond to reviewer Cyuw's comments and questions below.
> # Weakness: The performance are not convincing
> **Performance improvement**
> The most comparable method with SDAC in the safety gym is WCSAC.
> However, as mentioned in Q1 of the general response, it can be said that the proposed method is better than WCSAC regarding risk-averse property and multi-constraint setting.
> Please see Q1 of the general response.
> Also, in the locomotion tasks, SDAC shows the highest reward sum and meets constraints the fastest in all tasks.
> Thus, it can be seen that the performance is improved in the risk-averse and multi-constraint settings.
>
> **Constraint satisfaction**
> In the safety gym tasks, since the optimal policy exists on the boundary of the constraint, the training results exist at the boundary rather than rigidly satisfying the constraints.
> Nevertheless, a risk-averse constraint can be used by setting $\alpha=0.25$ to meet the constraint solidly.
> Figure 3.a shows that SDAC with $\alpha=0.25$ has the lowest cost (showing solid constraint satisfaction) and is in the middle or upper ranks of the reward sum.
>
> **Convergence to feasible policies**
> The convergence speed can be measured by the shortest step satisfying all constraints.
> The proposed method satisfied constraints at 1.5e6 step in Cassie, 0.25e6 step in Laikago, and 0.2e6 step in Cheetah, but the second-best method satisfied constraints at 3.5e6 step, 0.7e6 step, and 0.25e6 step in each task.
> Thus, the proposed method converged to the feasible policies the fastest.
> # Weakness: Two contributions are not related
> As mentioned in the introduction, if the estimation bias of critics becomes large, the policy can be trained to be overly conservative or risky.
> If the number of constraints increases, the possibility of updating the policy in a wrong direction rises exponentially.
> Thus, the importance of reducing the bias in the multiple constraint setting increases.
> To this end, we contribute to reducing the estimation bias of critics while handling multiple constraints.
> # Q1
> In the implementation, the policy takes random actions during the initial 10000 steps, like other SAC-based algorithms.
> As a result, low rewards were received, making the learning graphs discontinuous.
> We have attached an enlarged graph for the initial steps to the pdf of the general response.
> Please refer to the pdf.
> # Q2
> Lemma A.2 in the Appendix shows that the solution of equation (9) always exists under assumptions that the constraints are convex and the feasible policy set is not empty (see the assumptions introduced in Theorem 3.1).
> Therefore, there is no conflict between constraints due to the existence of the solution.
> # Q3
> In Q2 of the general response, the worst-case time to meet all constraints of our method is expressed as:
> $$\frac{D_\mathrm{KL}}{\beta(1-\gamma)(\zeta-\frac{2\beta C_\mathrm{max}}{(1-\gamma)^2})\mathbb{E}\_t[\sum_i\lambda^t_i/W_t]}.$$
> The worst-case time of the naive method is the same as the above equation, except for the $\lambda$ part.
> In the naive method, $\lambda^t$ is an one-hot vector where $i$-th value, $\lambda_i^t$, is one only for corresponding to the randomly selected constraint.
> In other words, only $\mathbb{E}\_t[\sum_i\lambda_i^t/W_t]=\mathbb{E}\_t[\sum_i\lambda_i^t/||\sum_i\lambda_i^tA\_{C_k}^t||\_2]$ is different.
> Let us assume that the advantage vector follows a normal distribution.
> Then, the variance of $\sum_i\lambda_i^tA_{C_k}^t$ is smaller for lambdas with distributed values than for one-hot values.
> Then, the reciprocal of the 2-norm becomes larger, resulting in a decrease in the worst-case time.
> From this, the gradient integration has a benefit over the naive method as it reduces the variance of the advantage vector.
> Of course, we cannot officially say that the worst-case time of the proposed method is smaller than the naive method because the advantage vector does not follow the normal distribution. Still, we can deliver our insight on the benefit of gradient integration.
> Also, the proposed method shows better results than the naive method in the multi-constrained experiments of Section 5.2.
> In Figure 4.d, the proposed method takes 1.5e6 steps to meet all the constraints, but the naive method takes about 2.5e6 steps.
> Also, the naive method shows an unstable training as some constraints are violated even after the 2.5e6 step.
> # Q4
> As the reviewer commented, TRPO is computationally heavy, but in safe RL, it can be said that TRPO-based policy updates are more effective than first-order (or gradient descent) updates.
> If a policy is updated by the first-order method, the policy can change rapidly, resulting in the constraints being violated suddenly, but TRPO-based update allows more stable learning because the values of the constraints do not change significantly.
> For this reason, many safe RL algorithms are based on TRPO [R1, R2].
>
> Second, as the reviewer mentioned, the policy can be updated using weighted sum of the cost critics to deal with multiple constraints.
> This method exactly matches the Lagrangian method of updating the policy to minimize the weighted sum of the Lagrange multipliers and cost critics.
> However, since these methods usually update the policy and weights simultaneously, the policy update direction can easily fluctuate.
> In contrast, due to the strong duality of the QP problem, the solution of equation (9) can be expressed as $\sum_k \lambda_k H^{-1}g_k$, so the gradient integration we propose can also be seen as a weighted sum method.
> Unlike the Lagrangian methods, it automatically calculates the appropriate weight by solving the QP problem at each gradient step, so it has the advantage of being able to find the policy update direction more stably.
> # **References**
> [R1] Achiam et al. "Constrained policy optimization." ICML, 2017.
>
> [R2] Xu et al. "Crpo: A new approach for safe reinforcement learning with convergence guarantee." ICML, 2021.

---

> > ### Comment · Reviewer_Cyuw · 2023-08-18
> >
> > Thanks for the reply. The reply fully addressed my problems in the comments. I decided to increase my score to 6.
> >
> > Things prevent me from further improving my score is
> > (1) The limitations of trust-region-based second-order algorithms also apply to this paper. Especially for multiple constraint algorithms which are highly likely without local convexity and feasibility, the assumptions might be too harsh.
> > (2) Either TRPO, TD(lambda), or CVaR is some existing techniques.

---

> > > ### Author Response · Authors · 2023-08-20
> > > **Response to Reviewer Cyuw**
> > >
> > > Thanks for your reply.
> > > We are glad that our answers can help you with the questions in the comments.

---

### Official Review · Reviewer_HJro · 2023-07-05

**Soundness:** 2 fair
**Presentation:** 3 good
**Contribution:** 3 good
**Rating:** 6
**Confidence:** 3

**Summary:**

This study focuses on multiple constraints setting in safe reinforcement learning, and an interesting method is proposed by leveraging gradient integration methods. Moreover, the feasibility of multi-constraint problems is addressed, TD distribution method is introduced to decrease the estimation bias.

**Strengths:**

1. Multiple constraints are considered.
2. The code is provided.
3. The writing quality is good.

**Weaknesses:**

1. Some papers are not investigated, e.g., [1] and [2]

[1] Gu, S., Yang, L., Du, Y., Chen, G., Walter, F., Wang, J., ... & Knoll, A. (2022). A review of safe reinforcement learning: Methods, theory and applications. arXiv preprint arXiv:2205.10330.

[2] Garcıa, J., & Fernández, F. (2015). A comprehensive survey on safe reinforcement learning. Journal of Machine Learning Research, 16(1), 1437-1480.

2. The experimental results do not convince me, as shown in Figure 3, other baselines also present better performance than this study, e.g., the WCSAC method.
3. Could you compare the method with PPO Lagrangian?

**Questions:**

1. Can we handle multiple constraints by averaging the multiple constraints as one constraint?
2. The essential contribution is to design the distributional critics with low biases. How about the distributional actors for estimating different types of constraints?
3. What is the difference between PCPO and this method? In PCPO, they also make projections to optimize reward and cost.
4. Why does the $J(\pi)$ like the formation in page 3, line 93?
5. Why do we need a Shannon entropy?
6. How does the study have Equations (4) and (5)? Could the study provide analysis to prove them?
7. Since the method claims the algorithm is efficient and computation efficiency is better than other baselines, could the study provide the sample complexity?

**Limitations:**

1. The cost is assumed as convex. However, in most cases, the constraints may be nonconvex.
2. The computation complexity and sample complexity should be provided to prove the effectiveness of this study.

---

> ### Author Rebuttal · Authors · 2023-08-07
>
> We thank reviewer HJro for the feedback and thorough review on our work.
> We respond to reviewer HJro's comments and questions below.
>
> # Weakness: Some papers are not investigated.
>
> I will add the two papers mentioned by the reviewer to the safe RL part of the related work section.
>
> # Weakness: Experimental results in Figure 3.
>
> As mentioned in Q1 of the general response, it can be observed that SDAC is better than WCSAC in terms of risk-averse property and multi-constraint handling.
> Please see the general response.
>
> # Weakness: Comparison with PPO-Lagrangian.
>
> We completed the experiments of PPO-Lagrangian (PPO-L) in the Safety gym and Cassie task, and the experimental results can be found in the attached pdf of the global response.
> Since PPO-L is an on-policy algorithm, it can be confirmed that the convergence speeds are slow compared to other algorithms.
> Also, like other lagrangian methods, PPO-L has the disadvantage of unstable training because multipliers and policies are updated simultaneously.
>
> # Q1: Merging multiple constraints into a single constraint.
>
> It is difficult to handle multiple constraints as a single constraint by averaging because the feasible regions of the multiple constraints, $F_k(\pi) \leq d_k$ $\forall k$, and the averaged constraint, $\sum_kF_k(\pi)/K \leq \sum_k d_k/K$, are quite different.
> Matching the feasible region using a single constraint requires designing a new integrated cost function rather than taking the average. Still, as mentioned in the introduction as a drawback of RL, this process is time-consuming.
>
> # Q2: How about the distributional actors for estimating different types of constraints?
>
> SDAC can be extended to use other types of constraints using the distributional critic, but it is essential to find the upper bound of the constraint to apply the trust region method.
> No study has yet derived the upper bound of risk-based constraints except for the mean-std constraint, so we can deal with this extension as future work.
>
> # Q3: Difference from PCPO.
>
> PCPO only deals with a single constraint but can be extended to multiple constraint settings.
> Nevertheless, the most significant difference from our method is that PCPO does not guarantee that the policy will converge to the feasible region for the infeasible starting case, but we can.
> However, we did not cite the PCPO paper, so we will cite this paper.
>
> # Q4: Why does the $J(\pi)$ like the formation in page 3?
>
> If we replace the $Z_R^\pi$ with the definition in $J(\pi)$, it becomes $\sum_t \gamma^t(R(s_t, a_t, s_{t+1}) + H(\pi(\cdot|s_t)))$, equivalent to the entropy-regularized RL problem [R1].
> We will explain why the entropy term is added in the answer to the next question.
>
> # Q5: Why do we need a Shannon entropy?
>
> The reason for adding the entropy term is to improve the exploration as in many existing RL papers, including Soft Actor-Critic (SAC) [R1], and it is not a required term.
> We additionally discussed that the trust region method can still be used even if an entropy term is added in Appendix A.7.
>
> # Q6: How does the study have Equations (4) and (5)?
>
> Since $\mathrm{Std}[Z_{C_k}^\pi]=\sqrt{\mathbb{E}[(Z_{C_k}^\pi)^2] - \mathbb{E}[Z_{C_k}^\pi]^2} = \sqrt{J_{S_k}(\pi) - J_{C_k}(\pi)^2}$, we can achieve equation (4) by substituting the Std term in equation (2).
> Equation (5) defines the surrogate functions for the objective and mean-std constraints.
> The condition for the surrogate function in the trust region method is that the difference between the original function and the surrogate should be bounded by the trust region size.
> For the objective's surrogate, we derive the bound in Appendix A.7, and the bound for the mean-std constraint's surrogate is derived in [R2].
>
> # Q7: Sample complexity.
>
> We analyzed the sample complexity of the proposed method in Q2 of the general response.
> Please refer to the general response.
>
> # References
>
> [R1] Haarnoja, Tuomas, et al. "Soft actor-critic: Off-policy maximum entropy deep reinforcement learning with a stochastic actor." International conference on machine learning. PMLR, 2018.
>
> [R2] Kim, Dohyeong, and Songhwai Oh. "Efficient off-policy safe reinforcement learning using trust region conditional value at risk." IEEE Robotics and Automation Letters 7.3 (2022): 7644-7651.

---

> > ### Comment · Reviewer_HJro · 2023-08-22
> >
> > Thanks for your response, I plan to upgrade the score.

---

> > > ### Author Response · Authors · 2023-08-22
> > > **Response to Reviewer HJro**
> > >
> > > We appreciate your thoughtful feedback.
> > > Thank you for raising your score.

---

### Official Review · Reviewer_dXag · 2023-07-05

**Soundness:** 3 good
**Presentation:** 2 fair
**Contribution:** 3 good
**Rating:** 6
**Confidence:** 3

**Summary:**

The paper tries to address the problem of safe RL with multiple constraints with a safe distributional actor-critic (SDAC) approach. The approach includes a gradient integration method to manage the infeasibility issues in multi-constrained problems and a TD$(\lambda)$ target distribution to estimate risk-averse constraints with low biases. Experimental results show that the proposed approach outperforms the baselines in both single- and multi-constrained tasks.


**Strengths:**

Safe Rl with multiple constraints is an important problem. The presented approach uses a gradient integration method to address the infeasibility issues of the trust-region-based algorithms with theoretical guarantees and proposed a TD$(\lambda)$ loss to reduce the estimation bias of the critics. Sufficient experimental results and ablation studies are provided to support the claims of the paper.


**Weaknesses:**

1. The writing of the paper can be improved. There are little background information and intuitions introduced, which makes the paper hard to read. For example, In Section 2 when introducing the trust-region method with a mean-std constraint, it would be better if the authors can introduce the intuitions behind the equations instead of just putting the equations there, because [Kim and Oh, 2022a] is not a very well-known paper. Also in Section 3.1, little information is provided about the intuition of equation (9).

2. See “Questions”.


**Questions:**

1. Seems the performance of the algorithm depends a lot on $\alpha$ and $\lambda$, then how can we choose them in the experiments?

2. Since the algorithm is designed to deal with multiple constraints, could the proposed algorithm deal with constraints with different priorities, for example, collisions with humans should be avoided prior to collisions with obstacles?

3. In Figure 3(a), WCSAC with both $\alpha=0.25$ and $\alpha=1.0$ satisfies the cost threshold in all environments while in some environments it also has higher rewards compared with SDAC. Could the authors provide some intuition about these observations?


**Limitations:**

The authors discuss the limitations well in Section 6.

---

> ### Author Rebuttal · Authors · 2023-08-07
>
> We thank reviewer dXag for the feedback and thorough review on our work.
> We respond to reviewer dXag's suggestions and comments below.
>
> # Weakness: The writing of the paper can be improved.
>
> As the reviewer commented, in constructing the subproblem in equation (6), many parts are enumerated without formal meaning, making it difficult to read.
> We will supplement the explanation of why the subproblem is established and what is necessary to establish it (the square value and surrogate functions).
> In addition, we attached Figure 1 to improve the understanding of equation (9), but the intuition about why the constraints should be truncated to the trust region was missing.
> The reason for truncating is to make the gradient integration method invariant to the gradient scale.
> Otherwise, a dominant policy gradient may form for constraints with large gradient scales.
> We will also supplement this explanation.
>
> # Q1: How can we choose $\lambda$ and $\alpha$ in the experiments?
>
> $\lambda$ adjusts the balancing between the bias and variance of the critics.
> We have experimented with various values of $\lambda$ in Section 5.3 and Appendix A.5 and found that the value between 0.9 and 1.0 is good as in other TD($\lambda$)-based papers [R1].
> Thus, we recommend setting $\lambda$ between 0.9 and 1.0.
> To give an intuition for setting $\alpha$, we can assume that the returns follow a Gaussian distribution.
> Then, the mean-std constraint becomes setting the probability that the return will be lower than the threshold.
> Thus, if you want to the cost return to be smaller than the threshold with a probability of $p$, you can find $\alpha$ as $p = \Phi(\phi(\Phi^{-1}(\alpha))/\alpha)$, where $\Phi$ and $\phi$ are the cdf and pdf of the standard normal distribution.
> Also, we wrote the tips for hyperparameter tuning in Appendix B, so please refer to it for the details.
>
> # Q2: Dealing with constraints with different priorities.
>
> Priority can be set indirectly through the thresholds of constraints.
> For the reviewer's example case, it can be implemented by setting the constraint for collision with people and the constraint for collision with obstacles separately and setting the threshold of the people constraint lower than the obstacle constraint.
> By giving different thresholds, we can generate a policy in which the probability of collision with a person is lower than the probability of collision with an obstacle.
> However, in decision-making situations where you have to choose between a collision with a person and an obstacle, it is challenging to always prioritize the choice of collision with an obstacle over a person.
>
> # Q3: Comparison of SDAC and WCSAC results in Figure 3.
>
> We presume that since WCSAC based on SAC [R2] takes a more significant number of gradient steps than SDAC based on TRPO [R3] (about 1000 times more), WCSAC performed better than SDAC on easy tasks. (All algorithms have high scores in the car-goal and point-button tasks, indicating that the tasks are easy.)
> However, as mentioned in Q1 of the global response, WCSAC does not significantly change the number of constraint violations even when the $\alpha$ value is adjusted, so it can be seen that SDAC is better in terms of the risk-averse property.
>
> # References
>
> [R1] Schulman, John, et al. "High-dimensional continuous control using generalized advantage estimation." arXiv preprint arXiv:1506.02438 (2015).
>
> [R2] Haarnoja, Tuomas, et al. "Soft actor-critic: Off-policy maximum entropy deep reinforcement learning with a stochastic actor." International conference on machine learning. PMLR, 2018.
>
> [R3] Schulman, John, et al. "Trust region policy optimization." International conference on machine learning. PMLR, 2015.

---

> > ### Comment · Reviewer_dXag · 2023-08-21
> >
> > I would like to thank the authors for their reply. I will keep my original score.

---

> > > ### Author Response · Authors · 2023-08-22
> > > **Response to Reviewer dXag**
> > >
> > > Thanks for your response.
> > > We appreciate for the positive review.

---

### Official Review · Reviewer_A8gA · 2023-07-07

**Soundness:** 3 good
**Presentation:** 3 good
**Contribution:** 3 good
**Rating:** 6
**Confidence:** 4

**Summary:**

The paper presents a safe reinforcement learning (RL) algorithm called SDAC for handling multiple constraints in safety-critical robotic tasks. SDAC incorporates risk-averse constraints and makes two key contributions: a gradient integration method for handling infeasibility issues and a TD($\lambda$) target distribution for estimating risk-averse constraints. Experimental results show that SDAC outperforms safe RL baselines, achieving fewer constraint violations and faster constraint satisfaction.


**Strengths:**

In general, the work is solid and contributes quite some novel ideas for safe RL.

- The formulation of safe RL problem with multiple constraints is very important for real applications.
- The infeasibility issues in constrained optimization are explicitly considered.
- The risk-averse safety measures are practical to safety-critical problems.



**Weaknesses:**

- While the measures have low bias, it is still important to consider the bias when making decisions.
- The presentation of the technical details, particularly the quantile regression part, is challenging to comprehend. It might be beneficial to refer to Dabney's paper, which provides a clearer explanation of the related content.
- From the empirical analysis, the proposed method cannot ensure safety during the early stage of training, despite the theoretical guarantees.


**Questions:**

- What is the purpose of adding a policy entropy term in line 91? Is it necessary, given the use of trust-region methods, and how is it maximized?
- Is there an implicit assumption that the distribution is Gaussian considering the risk measure in Equation 2?
- What is the reason for balancing bias and variance? Even though the bias is low, are there still some negative effects? How can we mitigate them, especially in the context of risk?
- How will the algorithm work if the constraints completely conflict with each other?
- When the cost signal is binary, multiple constraints can be taken as one constraint. Then, a method like WCSAC or CPO can also be used. In this case, what are the main advantages of SDAC?

**Limitations:**

Yes, the limitations are clearly stated at the end of the paper.

---

> ### Author Rebuttal · Authors · 2023-08-07
>
> We thank reviewer A8gA for the feedback and thorough review on our work.
> We appreciate that the reviewer commented that the proposed method is novel and solid.
> We respond to the reviewer's comments and questions below.
>
> # **Weakness: Presentation of the quantile regression can be improved.**
> As the reviewer commented, we found that the explanation of the quantile regression part can be difficult to follow due to the omitting intermediate steps in equations (11) and (12).
> We will include the derivation of each equation in detail and explain the intuition of each equation.
>
> # **Weakness: Safety violation during early stage of training.**
> As we mentioned in the introduction, infeasible starting cases are typical if there is no prior knowledge.
> In our experiments, we randomly initialize the policy networks resulting in the infeasible starting case.
> We can use the gradient integration method to recover the policy to the feasible region to handle the infeasible starting case, which is one of the main contributions of our paper.
>
> # Q1
> The reason for adding the entropy term is to improve the exploration as in many existing RL papers, including Soft Actor-Critic (SAC) [R1], and it is not a required term.
> We additionally discussed that the trust region method can still be used even if an entropy term is added in Appendix A.7.
> To maximize the entropy term, we can construct the objective value as $Q(s, a) - \beta\log{\pi(a|s)}$ and update the policy using backpropagation and reparameterization trick (please see SAC paper [R1]).
>
> # Q2
> There is no assumption of Gaussian distribution in equation (2).
> Nevertheless, if we assume the return follows a Gaussian distribution, equation (2) becomes equivalent to conditional value-at-risk (CVaR), a well-known risk measure in finance.
>
> # Q3
> In safe RL, since the safety of the policy is evaluated by the cost critics, a policy with the desired safety level can be generated only by training the critics with low biases.
> However, since the variance increases as the bias decreases in the TD($\lambda$) approach, the reward performance may decrease at $\lambda=1$, as in the ablation study (Figure 3.b).
> For this reason, balancing the variance and bias is essential.
> In order to mitigate balancing, it is important to set $\lambda$, and according to our ablation results, it is good to set a value between 0.9 and 1.0.
> In the context of risk, we can also increase the training batch size since risk measurement is more sensitive to sample noise.
>
> # Q4
> If the constraints conflict with each other, the solution of equation (9) does not exist.
> Then the policy is not updated, so the conflict will make the policy get stuck at one point.
> However, according to Lemma A.2 in the Appendix of our paper, the solution of equation (9) always exists under our assumption that the constraints are convex and the policy set that satisfies the constraints is not empty.
> Also, in our experiments, no conflict occurred because the gradients of constraints was calculated stochastically. (Since the dimension of the gradient is much larger than the number of constraints, there is little chance of conflicts in the case of stochastic gradient calculation.)
>
> # Q5
> As the reviewer commented, we can merge multiple constraints into a single constraint by the "and" operator in case of each cost function is binary.
> However, if the cost function is integrated in this way, it is difficult to establish a unified safety level (or threshold) that corresponds to the original constraint settings.
> We can give an example from the locomotion tasks, which have three constraints regarding body balance, CoM height, and feet timing.
> Since the body balance and the CoM height significantly affect the robot's stability, it is necessary to increase the safety level of the corresponding constraints.
> However, the constraint on the feet timing is to prevent the robot from standing still and has little relation to safety.
> Thus, if the level of this constraint is increased, the robot control may become unstable.
> Even though the integrated cost function can be defined as the result of the "and" operation of all cost functions, finding a proper safety level (or threshold) is tricky.
> For this reason, SDAC with multiple constraints has an advantage that can set the safety levels of each constraint independently.
>
> # References
>
> [R1] Haarnoja, Tuomas, et al. "Soft actor-critic: Off-policy maximum entropy deep reinforcement learning with a stochastic actor." International conference on machine learning. PMLR, 2018.

---

> > ### Comment · Reviewer_A8gA · 2023-08-21
> >
> > I would like to thank the authors for addressing my questions. I am keeping a positive recommendation for the paper and have no further questions.

---

> > > ### Author Response · Authors · 2023-08-21
> > > **Response to Reviewer A8gA**
> > >
> > > Thank you for your response.
> > > We appreciate for the positive feedback on our paper.

---

### Author Rebuttal · Authors · 2023-08-07

# General response
We thank all reviewers for their valuable comments and suggestions.
In the following, we respond to the comments on comparison with WCSAC and the sample complexity.
### **Q1. Comparison with WCSAC.**
We first examine the experiments in Section 5.1.
For $\alpha=1.0$ (risk-neutral constraint setting), SDAC achieves higher reward sums in the point-goal and car-button tasks, whereas WCSAC shows higher in the other two tasks.
The same results are observed at $\alpha=0.25$ (risk-averse setting).
However, risk-averse properties should also be taken into account in interpreting the results.
WCSAC uses risk-averse constraints like SDAC, so it should be able to significantly reduce the number of constraint violations at $\alpha=0.25$, but it failed to do so. (For an analysis of this reason, please refer to Section 5.1 of the paper.)
From this observation, SDAC has better risk-averse properties than WCSAC.
Moreover, in the multiple constraints setting (Section 5.2), whether in terms of reward or constraint satisfaction, SDAC outperforms WCSAC.
Hence, it can be concluded that SDAC is better than WCSAC in both the risk-averse and multi-constraint settings.
### **Q2. Sample complexity.**
To analyze the sample complexity, we consider tabular MDPs and use softmax policy parameterization (see [R1]).
It is challenging to analyze the sample complexity of TRPO-based methods since their stepsize is not fixed, so using natural policy gradient (NPG) methods is common, as done in [R1].
Thus, we introduce the NPG version of our method as follows.
1. (Recovery) If the constraints are not satisfied at $t$ step, we take a recovery step using the gradient integration as: $g^*=\mathrm{argmin}_g\frac{1}{2}g^THg$ s.t. $g_k^Tg+c_k\leq 0$ $\forall k\in$ {$k|F_k(\pi_t;\alpha)>d_k$}, $\psi\_{t+1}=\psi_t+\beta g^*/||g^*||\_2$ ,where $\beta$ is a stepsize, and other notations are the same as in the paper.
2. (Normal) If the constraints are satisfied at $t$ step, we store $t$ in a set $\mathcal{N}$ and maximize the objective as: $\psi_{t+1} = \psi_{t} + \beta A_R^{\pi_t}/||A_R^{\pi_t}||_2,$ where $A_R^{\pi_t}$ is the vector expression of the advantage function $\in \mathbb{R}^{|S| |A|}$. Derivation of the gradient can be referred to [R1].

According to the Lemma A.2 in our paper, $g^*$ in the recovery step always exists and can be expressed as: $g^* = \sum_k \lambda_k H^{-1}g_k = \sum_k \lambda_k A_{C_k}^\pi$, where $\lambda_k \geq 0$.

#### **Analysis of worst-case time to satisfy all constraints**
During the recovery step, the policy can be expressed as:
$$\psi\_{t+1}=\psi_t-\beta\sum_k\lambda^t_kA\_{C_k}^t/W_t,\ W_t:=||\sum_k\lambda^t_kA\_{C_k}^t||\_2,\ \pi\_{t+1}(a|s)=\pi_t(a|s)\exp{\left(-\frac{\beta}{W_t}\sum_k\lambda^t_kA\_{C_k}^t(s, a)\right)}/Z_t(s),$$
where $Z_t(s)$ is a normalization factor.
We can get the followings by using the Lemma 6 in [R1]:
$$\sum_k\lambda^t_k(F_k(\pi\_{t+1})-F_k(\pi_t))/W_t=\frac{1}{1 - \gamma}\mathbb{E}\_{s\sim\nu_\rho}\left[\sum_a\pi\_{t+1}(a|s)\sum_k\lambda^t_kA\_{C_k}^t(s, a)/W_t\right]\leq-\frac{1}{\beta}\mathbb{E}\_{s\sim\rho}\left[\log{Z_t(s)}\right],$$
where we abbreviate $F_k(\pi_t;\alpha)$ as $F_k(\pi_t)$.
We can also get the followings by using the Lemma 7 in [R1]:
$$\sum_k\lambda^t_k(F_k(\pi_*)-F_k(\pi_t))/W_t\geq-\frac{1}{\beta(1-\gamma)}\mathbb{E}\_{s\sim\nu_*}\left[D_\mathrm{KL}(\pi_*||\pi_t)-D_\mathrm{KL}(\pi_*||\pi\_{t+1})\right]-\sum_k\lambda^t_k\frac{2\beta C_\mathrm{max}}{(1-\gamma)^2W_t},$$
where $\pi_*$ is an optimal policy, $C_\mathrm{max}$ is the maximum value of reward and costs.
If $\lambda^t_k > 0$, $F_k(\pi_{t}) - F_k(\pi_*) > \zeta$.
Thus, $\sum_k \lambda^t_k(F_k(\pi_*) - F_k(\pi_{t})) \leq -\zeta\sum_k \lambda^t_k$.
If the policy does not satisfy the constraints until $T$ step, the following inequality holds by summing the above inequalities from $t=0$ to $T$:
$$\beta(1-\gamma)(\zeta-\frac{2\beta C_\mathrm{max}}{(1-\gamma)^2})\sum_{t=0}^T\sum_i\lambda^t_i/W_t\leq\mathbb{E}\_{s\sim\nu_*}\left[D_\mathrm{KL}(\pi_*||\pi_0)\right].$$
Let denote $\frac{1}{T}\sum_{t=0}^T \sum_i \lambda^t_i / W_t$ as $\mathbb{E}\_t[\sum_i\lambda^t_i/W_t]$, and we can get $W_t=||\sum_k \lambda^t_kA\_{C_k}^t||\_2\leq\sum_k\lambda^t_k2|S||A|C_\mathrm{max}/(1-\gamma)$.
Then, the maximum $T$ can be expressed as:
$$
T\leq\frac{D_\mathrm{KL}}{\beta(1-\gamma)(\zeta-\frac{2\beta C_\mathrm{max}}{(1-\gamma)^2})\mathbb{E}\_t[\sum_i\lambda^t_i/W_t]}\leq\frac{2|S||A|C_\mathrm{max}D_\mathrm{KL}}{\beta (1-\gamma)^2\zeta-2\beta^2C_\mathrm{max}}=:T_\mathrm{max},
$$
where we abbreviate $\mathbb{E}\_{s\sim\nu_*}\left[D_\mathrm{KL}(\pi_*||\pi_0)\right]$ as $D_\mathrm{KL}$.
Finally, the policy can reach the feasible region within $T_\mathrm{max}$ steps.

#### **Convergence rate**
For the normal step, we also can get the following inequality as done in the recovery step:
$$
\frac{(1-\gamma)^2\beta}{2C_\mathrm{max}}(J(\pi_*)-J(\pi_t))-\beta^2\leq|S||A|\mathbb{E}\_{s\sim\nu_*}\left[D_\mathrm{KL}(\pi_*||\pi_t)-D_\mathrm{KL}(\pi_*||\pi\_{t+1})\right].
$$
If we sum up the above inequalities from $t=0$ to $T$, we can get the following inequality:
$$
\frac{(1-\gamma)^2\beta}{2C_\mathrm{max}}(\sum_{t\in\mathcal{N}} (J(\pi_*) - J(\pi_t)) + \zeta (T - |\mathcal{N}|)) \leq |S||A|D_\mathrm{KL} + \beta^2T
$$, and according to Lemma 9 in [R1], the set $\mathcal{N}$ is not empty if $T$ is large enough.
Then, if we schedule $\beta = 2C_\mathrm{max}\sqrt{|S||A|/T}$ and $\zeta=2(D_\mathrm{KL} + 4C_\mathrm{max}^2)\sqrt{|S||A|/T}$, we can get convergence rate as: $J(\pi_*)-\mathbb{E}\_{t\in\mathcal{N}}[J(\pi_t)] = 2\sqrt{|S||A|/T}(D_\mathrm{KL} + 4C_\mathrm{max}^2)/(1 - \gamma)^2 = \mathcal{O}(1/\sqrt{T})$.

### **References**
[R1] Xu, Tengyu, Yingbin Liang, and Guanghui Lan. "Crpo: A new approach for safe reinforcement learning with convergence guarantee." International Conference on Machine Learning. PMLR, 2021.

---

### Decision · Program_Chairs · 2023-09-21

**Decision:**

Accept (poster)

**Comment:**

There is a consensus among reviewers that this paper meets the threshold for acceptance. There is a broad view that the paper is tackling an interesting problem, and that dealing with multiple constraints and infeasibility are helpful in expanding the applicability of the contributions. Some drawbacks were also identified by reviewers in the original reviews and discussion. There is some disagreement among reviewers on the clarity of the paper, and some views that the assumptions required in the core theoretical results may be too strong to be applicable in certain scenarios (e.g positive lower bounds on Hessian eigenvalues of constraint functions). Overall the decision is to accept this paper to NeurIPS; I hope the the reviews and results of the author-reviewer discussion help the authors in preparing their camera-ready version.